# Marine phytoplankton functional types exhibit diverse responses to thermal change

S. I. Anderson [1,5✉], A. D. Barton[2], S. Clayton[3], S. Dutkiewicz [4] & T. A. Rynearson [1✉]

Marine phytoplankton generate half of global primary production, making them essential to ecosystem functioning and biogeochemical cycling. Though phytoplankton are phylogenetically diverse, studies rarely designate unique thermal traits to different taxa, resulting in coarse representations of phytoplankton thermal responses. Here we assessed phytoplankton functional responses to temperature using empirically derived thermal growth rates from four principal contributors to marine productivity: diatoms, dinoflagellates, cyanobacteria, and coccolithophores. Using modeled sea surface temperatures for 1950–1970 and 2080–2100, we explored potential alterations to each group's growth rates and geographical distribution under a future climate change scenario. Contrary to the commonly applied Eppley formulation, our data suggest phytoplankton functional types may be characterized by different temperature coefficients ($Q_{10}$), growth maxima thermal dependencies, and thermal ranges which would drive dissimilar responses to each degree of temperature change. These differences, when applied in response to global simulations of future temperature, result in taxon-specific projections of growth and geographic distribution, with low-latitude coccolithophores facing considerable decreases and cyanobacteria substantial increases in growth rates. These results suggest that the singular effect of changing temperature may alter phytoplankton global community structure, owing to the significant variability in thermal response between phytoplankton functional types.

[1] Graduate School of Oceanography, University of Rhode Island, Narragansett, RI, USA. [2] Scripps Institution of Oceanography and Section of Ecology, Behavior and Evolution, UC San Diego, La Jolla, CA, USA. [3] Department of Ocean and Earth Sciences, Old Dominion University, Norfolk, VA, USA. [4] Department of Earth, Atmospheric and Planetary Sciences, Massachusetts Institute of Technology, Cambridge, MA, USA. [5] Present address: Department of Earth, Atmospheric and Planetary Sciences, Massachusetts Institute of Technology, Cambridge, MA, USA. ✉email: siander@mit.edu; rynearson@uri.edu

Phytoplankton are the primary contributors to marine productivity[1], fixing roughly 45 gigatons of carbon each year[2]. Like other organisms, both terrestrial[3] and marine[4], phytoplankton are susceptible to climate change-driven warming, but constraining how their communities may change in a future ocean remains challenging, as phytoplankton comprise phylogenetically diverse organisms with disparate evolutionary backgrounds. As a collective group, warming will directly impact their metabolic rates, potentially altering global primary production[5]. The directionality of the productivity response, either positive or negative, will depend on the temperature sensitivity of phytoplankton[5–7] and the diversity of thermal niches they occupy[5,8,9]. However, universal thermal growth sensitivities are commonly applied across all phytoplankton taxa, resulting in coarse estimations of community structure and productivity both in present day and future oceans[5,9–11].

Though much work has been done to establish that thermal traits vary among phytoplankton species[12–16], the ecological implications of differing thermal responses have not been fully explored, despite their clear relevance for understanding phytoplankton community structure and productivity. For example, because phytoplankton species have variable thermal responses, changes in water temperature may lead to differential changes in growth rates and shifts in distribution[17]. Over time, disparities in the thermal response and adaptive migrations could alter community structure in the world's oceans[18–20], potentially resulting in future communities with species compositions that have no analog to the present day[21].

In this study, we assess the relative capacities of phytoplankton to cope with ocean warming and illustrate the implications of varied thermal traits on growth and geographic range. We begin by characterizing the thermal responses of key phytoplankton functional types (PFTs), and depart from previous studies and methodologies by defining each group's unique temperature-growth interaction. This insight is essential to deciphering a suite of global biogeochemical processes, as changes in PFT composition, resulting from differences in their thermal responses, can drive shifts in biogeochemical stability[2], carbon export efficiency[22], and nutrient cycling[23]. The PFTs we evaluate include four principal contributors to marine productivity: diatoms, dinoflagellates, coccolithophores, and cyanobacteria. We employ sea surface temperature projections from an ensemble of Earth System Models to assess how PFT growth and geographical range may be altered in a future ocean. Though a multitude of factors

(e.g. nutrient supplies, ocean acidification, irradiance) ultimately influence phytoplankton distributions and global primary productivity[24], we focus on the impact of temperature in a future world alone, as: (a) sea surface temperature is widely believed to play a key role in shaping phytoplankton physiology and community structure, and (b) confidence in ocean surface temperature projections at regional and larger scales over the coming century are high compared with other environmental factors such as light and nutrients[20,25], which are impacted by complex biological processes. Our results suggest PFTs may exhibit different temperature coefficients ($Q_{10}$), growth maxima thermal dependencies, and thermal ranges, which would drive disparate responses to ocean warming, with the potential to alter phytoplankton global community structure.

## Results and discussion

**Disparities in the thermal response.** With previous meta-analyses[15,26] as a starting point, we compiled and quality controlled growth rates from 243 marine phytoplankton strains comprising 3246 discrete growth rate measurements at a broad range of temperatures and locations (Table 1, Supplementary Fig. 1, "Methods") in order to characterize the thermal response of diatoms, dinoflagellates, coccolithophores, and cyanobacteria. For these analyses, we constrained the cyanobacteria to exclude diazotrophs, as they are subject to fundamentally different physiological processes and the data available is sparse ("Methods"). These modifications resulted in PFT compilations that differed by as much as 72% (23 newly added of 32 cyanobacteria strains; Supplementary Table 1) from previous work[15,26]. Thermal reaction norms, curves characterizing growth as a function of temperature, were fit to individual strain growth measurements using an adapted Norberg curve (Fig. 1)[18] and used to evaluate the growth maxima ($\mu_{max}$, Fig. 2a inset) for each strain. The thermally viable range of each strain was then assessed using a 20% thermal performance breadth, calculated as the temperature range where growth rates were at least 20% of the $\mu_{max}$ ($\mu_{20\%max}$) for that strain (Fig. 2a inset, "Methods"). Evaluating the range in this manner reduced biases from highly skewed thermal reaction norms, brought about by inadequately defined thermal minima (Fig. 1). Differences in the absolute change in performance were then evaluated for each strain by assessing the thermal reaction norm slope ascending to or descending from the $\mu_{max}$ to the $\mu_{20\%max}$ ($|\mu|/°C$, Fig. 2a inset). Reaction norm slopes below and above the

**Table 1 Functional group thermal equation coefficients and dependencies.**

| Functional group | Source | Environment | n | N | y-intercept | $Q_{10}$ | $E_a$ | $\mu_{max}$ 20 °C |
|---|---|---|---|---|---|---|---|---|
| All | Eppley[28] | Marine | ~130 | 162 | 0.59 | 1.88 | 0.41 | 2.09 |
| | Bissinger et al.[29] | Marine | 92 | 1501 | 0.81 | 1.88 | 0.41 | 2.86 |
| | Chen et al.[71] | Marine | NA | 1387 | 0.45 | 1.43 | 0.23 | 0.93 |
| | This study | Marine | 243 | 3246 | 0.85 | 1.46 | 0.24 | 1.82 |
| Coccolithophores | Buitenhuis et al.[80] | Marine | 6 | 30 | 0.22 | 1.7 | 0.34 | 0.64 |
| | This study | Marine | 30 | 202 | 0.74 | 1.42 | 0.23 | 1.50 |
| Cyanobacteria | Stawiarski et al.[81] | Marine | 3 | 59 | 0.02 | 4.9 | 1.02 | 0.55 |
| | Chen et al.[71] | Marine | 36* | NA | NA | 4.74 | 1.00 | NA |
| | Kremer et al.[15] | Fresh & Marine | 106* | 968 | 0.58 | 1.61 | 0.30 | 0.95 |
| | This study | Marine | 32 | 502 | 0.19 | 2.13 | 0.49 | 0.86 |
| Diatoms | Chen et al.[71] | Marine | 134 | NA | NA | 2.08 | 0.47 | NA |
| | Kremer et al.[15] | Fresh & Marine | 169 | 1858 | 1.16 | 1.61 | 0.30 | 1.91 |
| | This study | Marine | 135 | 1794 | 0.80 | 1.55 | 0.28 | 1.91 |
| Dinoflagellates | Kremer et al.[15] | Fresh & Marine | 50 | 577 | 0.39 | 1.61 | 0.30 | 1.00 |
| | This study | Marine | 46 | 748 | 0.29 | 1.67 | 0.33 | 0.81 |

For each functional group and study, the environment examined, the number of strains examined (*n*) and total number of discrete growth measurements used in curve fitting (*N*) are shown. The y-intercept, temperature coefficient ($Q_{10}$), activation energy ($E_a$), and growth maximum ($\mu_{max}$) at 20 °C are based on the exponential temperature dependency characterized in each study. Two studies included diazotrophic species (asterisks) but those were not incorporated into our analyses.

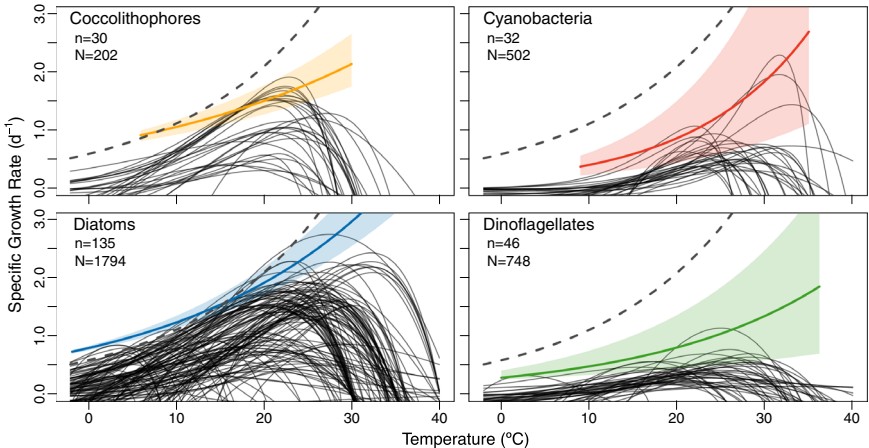

**Fig. 1 Thermal reaction norms for each phytoplankton functional type.** Modeled exponential curves (colored lines) were fit to measured growth rates (*N*) using a 99th quantile regression and compared to the widely-used Eppley curve[28], which assessed phytoplankton collectively (gray dashed line, same in each panel). Extent of modeled curves (colored lines) denote limits of data. The 95% confidence intervals (shading) were determined using Markov chain marginal bootstrapping[75] and are centered at the median. Thermal reaction norms (*n*) for each isolate characterized are shown in black. The cyanobacteria group does not include diazotrophic species ("Methods").

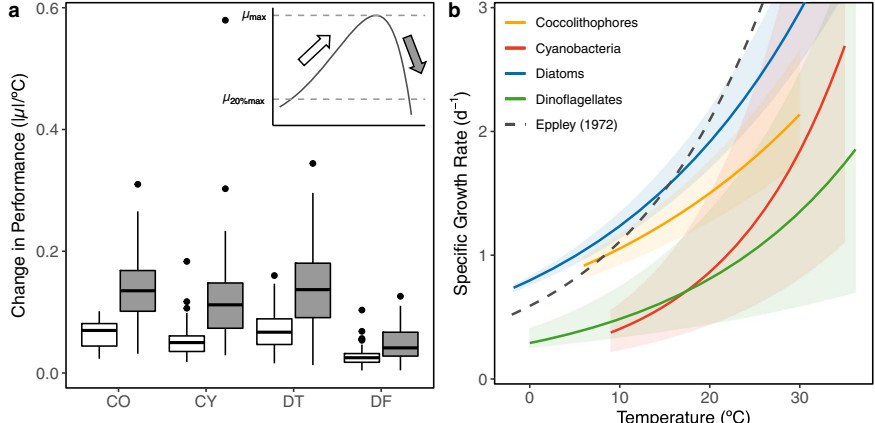

**Fig. 2 Comparison of thermal dependencies among phytoplankton functional types. a** Absolute change in performance for each PFT (Coccolithophores = CO, cyanobacteria = CY, diatoms = DT, dinoflagellates = DF), determined by analyzing the rate of change from 20% of the maximum growth rate ($\mu_{20\%max}$) to the $\mu_{max}$. Colors correspond to the rates approaching (white) or descending (gray) from the $\mu_{max}$. Box plots extend from the first to the third quartile, with a line denoting the median, whiskers extending to the greatest value within 1.5× interquartile range, and points displaying data outliers. Only strains for which reaction norms could be fit were used in calculations (CO = 30, CY = 32, DT = 135, DF = 46). All groups were negatively skewed resulting in greater rates of growth descent from the $\mu_{max}$. **b** Functional group maximum growth rates (modeled $\mu_{max}$) as a function of temperature fit using growth measurements according to Bissinger et al.[29] and compared with that from Eppley[28]. 95% confidence intervals (shading) were determined using Markov chain marginal bootstrapping[75]. Dinoflagellates were characterized by the smallest performance changes overall, as evidenced by their relatively flatter reaction norms (Fig. 1).

thermal optima ($T_{opt}$; temperature at $\mu_{max}$) were used to discern average PFT skewness, with a greater slope above the $T_{opt}$ indicative of a negatively skewed curve (Fig. 2a).

Our analyses identified several disparities among PFT thermal responses. We found that PFTs varied in their thermally viable range. For example, though cyanobacteria are known to have a narrower niche, and diatoms a wider niche, than other PFTs[26], consistent with our study (Supplementary Table 2), the positioning of their respective ranges also varied along the thermal gradient, with cyanobacteria not able to survive temperatures less than 9.5 °C (Supplementary Fig. 2A). Conversely, only 17% of the coccolithophores within our dataset were viable at temperatures greater than 30 °C, compared with >60% for all other groups examined. Variation in the thermal niche width was not significantly correlated with PFT sample size (Pearson's

correlation, df = 2, *p* = 0.2409) or absolute isolation latitude (Pearson's correlation, df = 225, *p* = 0.4507). While all PFTs were characterized by negatively skewed thermal reaction norms on average (Fig. 2a), consistent with previous findings[26], we found that strains from each PFT were also differentiated by the shapes of their thermal reaction norms. For instance, dinoflagellates had shallower slopes ascending to and descending from the $\mu_{max}$ (Fig. 2a), resulting in flatter curves (Fig. 1). This may be due to their significantly lower (Supplementary Table 2) and less variable $\mu_{max}$ than all other groups examined (Supplementary Fig. 2B), consistent with previous findings[16]. One explanation for dinoflagellates' differing thermal reaction norms may be their frequent exposure to thermal fluctuations resulting from their tendency to perform diel vertical migrations, a behavior that may have led to an evolutionary trade-off in which an enhanced thermal breadth

(viable range) developed at the expense of $\mu_{max}$[27]. These instances of variability, taken together, suggest PFT's may have dissimilar responses to thermal fluctuations and long-term climate change.

The temperature dependence of each PFT's maximum growth rate can be described using an exponential function[28,29] (Figs. 1 and 2b), which is commonly employed in Earth system models[7,10,30], satellite-based estimates of primary production[11], and growth rate standardizations[31]. However, most models apply the same temperature sensitivity (i.e., $Q_{10}$) across all phytoplankton types[10], which has the tendency to lead to over or underestimations of productivity depending on which PFT is the principal contributor in a given region[32]. To address this, we conducted the most comprehensive assessment of the marine phytoplankton thermal response to date, both in terms of the amount of data utilized and in our taxonomic differentiation of the thermal response (Table 1). To begin, we tested the null hypothesis that all functional groups are characterized by equivalent growth rates and thermal traits. We instead found PFTs to exhibit significantly different growth rates (Kruskal–Wallis, $p < 0.0001$; $\varepsilon^2 = 0.28$ [0.25, 0.31]) and $\mu_{max}$ (Supplementary Table 2; $\varepsilon^2 = 0.42$ [0.34, 0.50]). Based on these findings, we elected to model the $\mu_{max}$ as a function of temperature separately for each PFT, fitting exponential curves to thermal growth rates from each PFT. We utilized the 99th quantile regression method[29], which characterizes phytoplankton growth at the 99th percentile. This approach serves as a stronger estimator of group $\mu_{max}$ than methods that assess rates at the 50th percentile or those that use only a subset of the growth data in curve fitting ("Methods"). By characterizing the thermal response in this way, the temperature-PFT interaction could be numerically accounted for in each equation's exponent (Supplementary Table 3), as had not been done previously e.g., ref. 15, allowing for unique temperature coefficients ($Q_{10}$) to be calculated for each group. The $Q_{10}$ describes the thermal sensitivity of the PFT maximum growth rate (modeled $\mu_{max}$) with each 10 °C of temperature increase.

Our results differed from the well-established and widely-used Eppley curve[28], which characterized the thermal sensitivity of phytoplankton maximum growth rates ($Q_{10}$) as 1.88. We instead found the four functional types examined to have a collective $Q_{10}$ of 1.46 (Supplementary Fig. 3), closer to more recent evaluations[15]. Additionally, fitting exponential curves separately to each PFT resulted in widely varying $Q_{10}$ values and y-intercepts between PFTs, with cyanobacteria and dinoflagellates displaying significantly lower intercepts than either coccolithophores or diatoms ($\alpha = 0.05$; Fig. 1, Table 1, Supplementary Table 3). Cyanobacteria were also characterized by a significantly greater exponential slope than all other PFTs ($\alpha = 0.05$; Fig. 1, Table 1, Supplementary Table 3), which resulted in a higher $Q_{10}$ of 2.13, relative to Eppley, and contributed to the observed PFT-temperature interaction (Supplementary Table 4). Conversely, diatoms, coccolithophores, and dinoflagellates exhibited lower $Q_{10}$ values of 1.55, 1.42, and 1.67 respectively (Table 1). While several studies have suggested that Eppley's $Q_{10}$ of 1.88 may be an overestimation of phytoplankton thermal sensitivity[15,33], we instead propose that depending on the PFT, Eppley's value may be either an over- or underestimation, and taxonomically resolving the $Q_{10}$ can aid in accurately assessing the phytoplankton thermal response.

While thermal sensitivities describe the slope of the exponential function, the PFT maximum growth rate (modeled $\mu_{max}$) at a given temperature can provide a more absolute comparison between PFTs. For example, at 20 °C, diatoms and coccolithophores have the greatest modeled $\mu_{max}$ despite having the lowest $Q_{10}$ values (Table 1, Fig. 2b). This may indicate a higher competitive ability from a growth rate standpoint, even though

their thermal growth response, in terms of rate change, may be less. When looking across the full range of temperatures, diatoms exhibit the greatest modeled $\mu_{max}$ of all the PFTs examined (Fig. 2b) supporting the theory that they are r-strategists[34], maximizing growth when conditions are favorable. These findings also suggest that diatoms would dominate in favorable conditions (e.g., replete nutrients), which is consistent with several studies of diatom-coccolithophore competition in modern day oligotrophic regions[34,35] and in the paleo record[36]. The high $\mu_{max}$ values relative to other PFTs may also explain diatoms' propensity to excel in anomalous thermal conditions, such as marine heat waves[37,38], which have increased in frequency over the last century[39] and which are projected to intensify in the future[40].

Though the exact cause for these physiological differences remains unknown, the four PFTs represent distinct phyla with complex evolutionary histories spanning over 1 billion years, including two separate endosymbiotic events[41,42], which may underlie their unique thermal responses. Together, these findings support differentiating functional groups when assessing the phytoplankton thermal response and implementing growth rates in modeling studies.

**Thermal capacity across latitudes**. One of the many uses of thermal reaction norms is to evaluate thermal traits, such as the upper thermal limit ($T_{max}$) and the optimal temperature for growth ($T_{opt}$) (Fig. 3a). When an organism's environment is assessed in relation to these traits, one can begin to evaluate species-level thermal capacity, or tolerance for warming[43]. Here, we utilized three simple metrics, two of which are well-established[3,4,43,44], to characterize thermal capacity at the functional type level and illustrate varied traits in an ecological context. The thermal safety margin (TSM, Fig. 3b) describes an organism's thermal proximity to its $\mu_{max}$, using the difference between the organism's mean habitat temperature ($T_{hab}$) and its $T_{opt}$[3]. Second, we defined a new metric, termed the distance to the growth equivalence (DGE, Fig. 3c), which describes the distance (°C) to the temperature above $T_{opt}$ at which the growth rate is equivalent to that at the $T_{hab}$ ($T_{\mu equiv}$). This distance metric describes the warming that an organism can endure before its growth rate falls below that at the $T_{hab}$. The DGE and the TSM are similar, but the DGE accounts for growth potential above the $T_{opt}$, as phytoplankton are known to persist at sub-optimal temperatures[18]. While the DGE is only relevant for phytoplankton currently inhabiting temperatures below their $T_{opt}$, it allows for a more accurate depiction of organismal thermal capacity, showcasing the greater thermal range that can be tolerated before growth rates decline. Finally, the warming tolerance (WT, Fig. 3d), characterizes the amount of warming that can be tolerated before cell death occurs[3].

From these metrics, we find that while many mid-latitude and equatorial strains may be inhabiting temperatures above their $T_{opt}$ (Fig. 3b, negative TSM), the majority of strains are buffered from potential cell death by a substantial WT (Fig. 3d). Additionally, the DGE suggests more warming may be tolerated before growth decreases from that at its baseline $T_{hab}$ (Fig. 3c), which is critical to consider when evaluating processes such as primary production in a future ocean. While latitudinal trends in individual phytoplankton traits, such as $T_{opt}$ and $T_{max}$[26], may increase toward the equator, they do not scale at the rate of the $T_{hab}$ resulting in the observed hyperbolic tendency in the TSM and WT. This hyperbolic trend is consistent with other marine species[43], as well as terrestrial ectotherms[3], highlighting the potentially limited capacity of organisms to cope with warmer waters toward the equator. This suggests that in a future ocean,

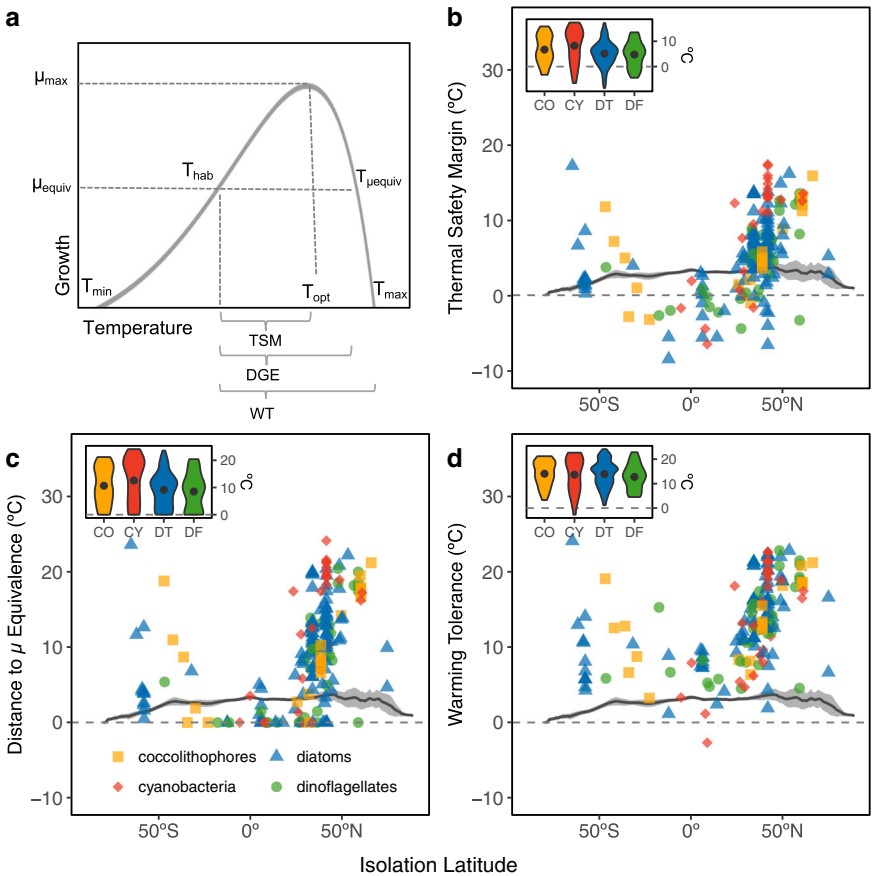

**Fig. 3 Thermal capacities for each functional type.** Thermal performance metrics (**a**), including thermal safety margin (**b**, TSM = $T_{opt}$ − $T_{hab}$), distance to the growth equivalence (**c**, DGE = $T_{\mu equiv}$ − $T_{hab}$) and warming tolerance (**d**, WT = $T_{max}$ − $T_{hab}$) are shown. An isolate existing below zero (dotted line) has already surpassed the given thermal capacity under historical, baseline conditions (1950–1970). For example, for the DGE (**c**), values of zero indicate individuals were isolated at temperatures at or above their $T_{opt}$. Gray solid line characterizes the amount of warming predicted for each latitude by 2100, with the shading representing longitudinal variation (sd). Violin plots within each figure (**b**–**d**) display the mean (points) and density distribution of thermal capacities for each functional group (Coccolithophores = CO, cyanobacteria = CY, diatoms = DT, dinoflagellates = DF). Only strains of known origin with good quality fit in $T_{max}$ were used in calculations (CO = 24, CY = 31, DT = 115, DF = 38).

temperature could alter the composition of existing phytoplankton communities. We found no significant differences in WT or DGE between PFTs (Kruskal–Wallis, $p = 0.5729$ & $p = 0.1075$, respectively; Fig. 3 insets), but cyanobacteria did have a significantly greater TSM from either diatoms or dinoflagellates (Dunn's test $p = 0.0568$ & $p = 0.0617$), indicating they often inhabit climates further below their optima.

**Changing rates and shifting ranges.** To further illustrate the implications of thermal differentiation between PFTs, we conducted a separate assessment of phytoplankton growth in the world's oceans, accounting for the global dispersal potential of phytoplankton[45], as well as the large temperature gradations that phytoplankton experience, and readily acclimate to due to phenotypic plasticity[46]. Assuming no limits to dispersal in phytoplankton, including those resulting from thermal boundaries, as supported by empirical observations[47] rather than modeling simulations[48], and under the premise that habitation is solely dependent on thermal viability and not dispersal or competitive ability ("Methods"), we evaluated the growth of every strain at all thermally viable locations across the global ocean. Viability was established where strain growth rates were greater than 20% of their strain-specific maximum growth rate (a 20% thermal performance breadth). Though many studies have arbitrarily set a higher threshold of 80%[49], previous work has indicated that

species commonly appear or even dominate in the field at temperatures corresponding to 20% of their $\mu_{max}$[12], supporting the use of a wider thermal breadth. We then used an ensemble mean of modeled SST projections from the Coupled Model Intercomparison Project phase 5 (CMIP5) under RCP8.5[50] to evaluate strain growth rates under historical (1950–1970) SST conditions and contrasted them with future projections (2080–2100) under anthropogenic warming. The median proportional growth change between the two time periods was then computed for each group and differences were averaged for each latitude ("Methods").

The majority of strains from each group are projected to experience proportional decreases in growth rates at low-latitudes and significant gains at mid-latitudes (Fig. 4a). However, the proportional change anticipated for each PFT is varied, suggesting community structure may be altered, as either PFTs shift in their relative competitive abilities as a group, or in the species which they comprise. Among the functional groups, low-latitude coccolithophores appear the most susceptible to rising temperatures, while cyanobacteria may fare the best, with significant growth increases expected at mid-latitudes (Fig. 4). Additionally, many new regions, such as the Norwegian Sea and the Gulf of Alaska, may become habitable to cyanobacteria under future thermal conditions (Fig. 4b), allowing for an average range expansion of about 6.5% or 18.8 million km² based on temperature alone. This cyanobacteria range expansion has the potential to alter community structure as cyanobacteria impose

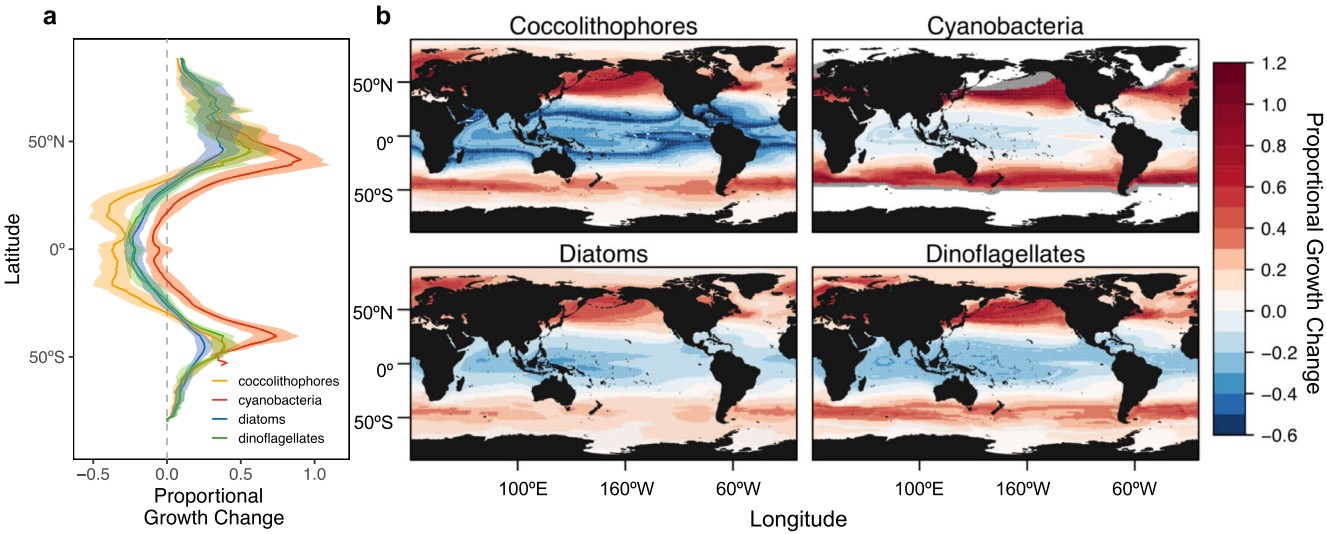

**Fig. 4 Change in proportional growth predicted between historical (1950–1970) and future (2080–2100) temperature regimes under the RCP 8.5 climate scenario. a** Latitudinal averages in proportional growth change are depicted for each functional group (lines) with shading representing longitudinal variation from the mean (sd). **b** Maps depict median proportional growth change from strains within each phytoplankton functional group. Strains were considered viable at a given location if their growth rates were at least 20% of their $\mu_{max}$. Gray area at cyanobacteria boundary extent indicates growth occurring in the future, but absent in the past (i.e., a range expansion).

competition in new regions. Our projections, coupled with those suggesting that phytoplankton with smaller cell sizes (such as cyanobacteria) will be advantaged in future lower nutrient conditions[5,51], make it plausible to presume that some species of larger cell size which currently dominate[52], may be displaced or outcompeted in the future. Like cyanobacteria, coccolithophores may also shift poleward[53], joining or replacing existing coccolithophore populations and potentially increasing their prevalence, as has been observed in the North Atlantic[54]. These immigrations have the potential to increase resource competition among PFTs and alter the current community structure at mid-latitudes[35].

Some of the greatest thermally-induced proportional changes in growth are predicted for low-latitude regions, which are already impacted by nutrient availability[52]. For example, on average, coccolithophores are estimated to experience as much as a 61% decrease in growth rates at low latitudes, with 100% of strains negatively impacted within 10° of the equator and 83% within 20° (Fig. 4). Given their key role in carbon cycling, this reduction in calcite-bearing coccolithophore growth could potentially alter seawater alkalinity[35,53]. These findings are consistent with trends illuminated using our DGE metric (Fig. 3c), which similarly found temperature increases at the equator to exceed the buffering capacity of most strains, resulting in negative growth changes. The severity of these projections may be lessened with more characterizations of low-latitude strains (Supplementary Fig. 1), but they also serve as base approximations, as they result from temperature alone and do not include other factors which are also likely to hinder or enhance net growth, such as increased stratification and resulting nutrient limitation[55], or interactions with other organisms. Taken together, future low-latitude regions will likely be characterized by very different environmental conditions and phytoplankton communities than those of today.

At mid and high latitudes, growth projections are the reverse, with proportional growth increases projected for all PFTs (Fig. 4). The Southern Ocean, for example, may see an average of a 7.2% proportional growth increase among diatoms, dinoflagellates, and coccolithophores. In addition, warming in the Southern Ocean

has the potential to increase iron supply through ice melt[56] and induce stratification[55], easing the nutrient and light limitation that currently impact the region. Together, these support the prediction that temperature could be the principal driver of biomass and productivity changes in the future Southern Ocean[5,56]. The North Atlantic may similarly be subject to a 21% proportional growth increase among all PFTs based on temperature alone. However, this projected increase may conversely be hindered by changing nutrient concentrations, as nitrogen limitation is presumed to intensify in a future ocean, potentially dominating over the thermal response[5]. Though our analyses do not account for additional factors that influence growth rates and the thermal response, including nutrients[13,57] and light[58], they provide a baseline of the impact that temperature alone could have on phytoplankton growth in the world's oceans.

In each region, we demonstrate that the direct effects of temperature will alter phytoplankton growth. Yet, our results also suggest that this thermal response will vary widely among PFTs, potentially reshaping phytoplankton communities. Models have previously predicted potential deviations in diversity with increasing SST[59], which similarly support the theory that future communities may not be analogous to those of the present day[21]. However, our focus solely on temperature allowed for taxonomic differentiations, elucidating some of the mechanisms that may drive these changes in diversity. Though evolutionary rates may increase with temperature[60], alleviating some of our predicted metabolic losses, the time-scale on which this could occur as well as the trade-offs associated with adaptation are only beginning to be explored in the phytoplankton[61–63]. Thus, our meta-analysis serves as a baseline for which the thermal response can be understood. Subsequent modeling efforts, satellite-based estimations of primary production[11], phytoplankton growth rate standardizations[31], or the many other ecological implementations of $Q_{10}$ will benefit from the incorporation of our newly characterized thermal dependencies for each PFT, allowing them to better decipher the network of ecological and biogeochemical processes impacted by the phytoplankton response to temperature.

## Methods

**Growth data compilation**. To assess the phytoplankton growth response to temperature, we aggregated thermal growth rates from four principal phytoplankton functional groups of biogeochemical relevance: coccolithophores (number of growth measurements ($N$) = 202), cyanobacteria ($N$ = 502), diatoms ($N$ = 1794), and dinoflagellates ($N$ = 748). We began with a previous thermal growth rate compilation by Thomas et al.[18,26] and added data published after 2012 (number of strains ($n$) = 59, $N$ = 883; Supplementary Table 1) which followed the same selection criteria outlined in Thomas et al.[18], with the following modifications; selection criteria were broadened to include growth rates measured at greater than 80 μmol photons m$^2$ s$^{-1}$ (rather than 100 μmol photons m$^2$s$^{-1}$) when day length equaled 24 h, allowing for the inclusion of more marine cyanobacteria, which had previously been relatively underrepresented in the data set. We also eliminated studies that exposed strains to fluctuating nutrient concentrations, as there was concern about the comparability of the resulting reaction norms[13,57]. Lastly, the cyanobacteria group was constrained to eliminate diazotrophic species, which are characterized by fundamentally different physiological processes, which could impact group characterizations. It should be noted that a separate analysis was conducted for the diazotrophs, as they are significant ecological contributors; however, the data was deemed insufficient and resulted in a high degree of uncertainty in subsequent analyses (Supplementary Fig. 4). Future work examining diazotroph thermal traits would be of great value to the scientific community. Additionally, dinoflagellate growth rates were verified to be autotrophically obtained (strains grown on medium only), but we cannot eliminate the possibility that some species may have ingested bacteria to augment their growth[64].

When growth data were not made available in spreadsheet form, GraphClick software (version 3.0.3)[65] was employed to digitize rate measurements from published figures. In total, our compilation included four functional groups comprising 243 strains and 3,246 discrete growth rate measurements from a broad range of temperatures and locations (Table 1, Supplementary Fig. 1)

Thermal reaction norms were used to describe each strain's thermal response following the equation presented in Thomas et al.[18], which was adapted from Norberg[66]. For strains compiled previously[18], parameters for thermal reaction norms were provided. For added strains, parameters were estimated using the maximum likelihood approach described in Thomas et al.[18] and the bbmle package[67] in R 4.0.2[68].

**Climate data**. To assess potential impacts of warming on phytoplankton metabolism, we utilized an ensemble mean of modeled sea surface temperature (SST) projections from the Coupled Model Intercomparison Project phase 5 (CMIP5), available at a 1.25° resolution, and presented in the Fifth Assessment Report of the Intergovernmental Panel on Climate Change[50]. This data was extracted from the Royal Netherlands Meteorological Institute Climate Explorer portal (http://climexp.knmi.nl). Projections for sea surface temperature (SST) warming were calculated between a baseline (1950−1970) and future (2080−2100) time period under Representative Concentration Pathway RCP8.5 (Supplementary Fig. 6), a worst case climate scenario that assumes increasing greenhouse gas emissions[69]. Warming was assessed at each latitude, resulting in a zonal mean and standard deviation. Habitat temperatures ($T_{hab}$) for strains of known origin were discerned by extracting SST values at strain isolation locations from the baseline period mean (1950–1970).

**Thermal dependencies**. For each functional group, the change in the maximum growth rate ($μ_{max}$) with temperature was characterized with an exponential function. This relationship was first described by Eppley[28], and a curve-fitting method was later standardized by Bissinger et al.[29]. We implemented the method outlined in Bissinger et al.[29] by fitting a 99th quantile regression to log-transformed growth rates from each functional group using the quantreg package in R[68,70] (Supplementary Fig. 5, Supplementary Table 3). Though results from quantile regression can be sensitive to the selected quantile (here 99th) and require substantial data inputs to estimate, we found this method preferential to other methodologies previously utilized to evaluate the $μ_{max}$-temperature relationship[15,71], such as the metabolic theory of ecology (MTE) or ordinary least squares (OLS) regression. Unlike the MTE, which incorporates only the $μ_{max}$ from each species for curve fitting[15], or the OLS regression, which uses only data below the thermal optima[72], the 99th quantile method utilizes all growth data available to assess thermal dependencies. Additionally, the 99th quantile describes community growth rates at the 99th percentile, providing a better estimation of maximum growth rates than the OLS method, which fits to the log-transformed data (proximate to the 50th quantile, median). Evaluating an extreme quantile also enables quantile regression to better capture changes in growth rate dispersion and variation in response to temperature, when compared to OLS regression[73]. Lastly, the 99th quantile method differs from MTE methodology in that it does not require assumptions regarding phytoplankton cell size, which is difficult to approximate in the phytoplankton, as it can vary significantly within species[74] and the size-scaling of growth is strongly dependent on temperature[12,72]. Thus, we decided against incorporating species size estimates that were unassociated with thermal growth experiments.

We fit 99th quantile regressions to each PFT's thermal growth rates separately. This stemmed from the findings that each PFT was characterized by significantly different growth rates (Kruskal–Wallis, $p < 0.0001$) and $μmax$ (Supplementary Table 2) contradicting the null hypothesis that all functional groups are characterized by equivalent thermal growth rates and thermal traits. We also compared two models of pooled thermal growth rates using the Akaike information criterion with correction for small sample size (AICc), which showed that including a temperature-PFT interaction resulted in a stronger model (Supplementary Table 4). This resulted in $μ_{max}$-temperature relationships characterized by the following equation:

$$μ_{max}(T) = a \cdot e^{b \cdot T} \qquad (1)$$

where the maximum growth rate of each functional group ($μ_{max}$) changes as a function of temperature ($T$). The y-intercept ($μ_{max}(0\,°C)$) is given by parameter $a$, and $b$ characterizes the rate at which the $μ_{max}$ increases with temperature. The 95% confidence interval for each curve was then estimated using a Markov chain marginal bootstrap[75] over 10,000 iterations[70] (Supplementary Table 3). This method lessens the computation required for bootstrapping by solving one-dimensional equations for multi-dimensional parameters[75], making it ideal for large data sets.

Exponential curves from our analyses were then utilized to estimate the temperature coefficient ($Q_{10}$) for each functional group. The $Q_{10}$ describes the rate at which the $μ_{max}$ changes with each 10 °C of temperature change, providing a valuable metric for metabolic capacity. In the past, functional groups have been included as a parameter in the exponential equation, resulting in a constant $b$ across functional groups, and thus a constant $Q_{10}$[15]. Due to our interest in differentiating between functional groups, we chose to fit exponential curves to each functional group separately, obtaining unique $Q_{10}$ values for each group. Activation energies ($E_a$) were then computed using each PFT's exponential equation or $Q_{10}$ following the methods in Kremer et al.[15].

**Static thermal capacity and statistical analyses**. There are several established metrics for assessing thermal capacity, including the thermal safety margin (TSM) and warming tolerance (WT)[3,4,43]. Each metric considers organismal thermal traits in relation to their habitat temperature to provide an estimate for the amount of warming that can be tolerated before performance decreases (TSM) or strains become non-viable (WT). They operate under the assumption that organisms are static, experiencing environmental temperatures corresponding to a single location. Though phytoplankton are subject to high dispersal[46,47], we utilized these metrics as touchstone assessments of thermal capacity. To calculate each metric, we estimated the thermal optima ($T_{opt}$) and thermal maxima ($T_{max}$) of each strain using their respective thermal reaction norms, as outlined previously[12,18,26]. The thermal maxima were then quality controlled according to Thomas et al.[26] to ensure validity.

Once trait values were quantified, metrics for thermal capacity were calculated for all strains of known origin with a well-characterized $T_{max}$[26] (coccolithophores = 24, cyanobacteria = 31, diatoms = 115, dinoflagellates = 38; Supplementary Fig. 1). Trait values were used to compute the TSM ($T_{opt} − T_{habitat}$) and the WT ($T_{max} − T_{habitat}$). Additionally, we defined a new metric termed the distance to the growth equivalence (DGE). While the TSM is often used to define the limit of performance, beyond which an organism's growth rate is hindered, we found this to be somewhat misleading as phytoplankton often perform below their capacity as they readily exist at temperatures below their $T_{opt}$[18]. Thus, we found it reasonable to postulate that phytoplankton could similarly operate at a reduced capacity beyond the $T_{opt}$. To account for this, we formulated the DGE, which describes the distance (°C) to the temperature at which growth is equivalent to that at the organism's mean habitat temperature ($T_{μequiv} − T_{habitat}$), but on the opposite side of the reaction norm (Fig. 3c). This characterizes the degree of warming that can be sustained before growth decreases below that in the organism's habitat. Variations in thermal metrics among PFTs were conducted using Kruskal–Wallis tests followed by Dunn's multiple comparison tests with an alpha of 0.05.

**Metabolic projections for the future**. In order to assess the implications of varied thermal responses among phytoplankton functional types, we estimated the proportional growth change that could be experienced between baseline (1950–1970) and future (2080–2100) thermal conditions (Supplementary Fig. 6). Strain reaction norms from each functional group were assessed in conjunction with modeled SST to estimate strain growth rates for each global grid cell (1.25° resolution). Strains were considered viable at a given location if their growth was at least 20% of their growth maxima ($μ_{20\%max}$). This percentage was based on observations of species presence at temperatures roughly corresponding to those at the $μ_{20\%max}$ or higher[12]. Proportional growth change was then calculated for grid cells in which strains were determined to be viable under both baseline and future conditions (($μ_{future} − μ_{past})/μ_{past}$). For each phytoplankton functional type, the proportional change was then computed using the median of individual strain results and depicted with a global map. While, by definition, some strains will fare better and some worse than the median, we found this to be the metric most suited for conveying how the majority of strains will respond, while being less sensitive than other metrics, like the maximum, to sample size and outliers. Additionally, our estimate of the median proportional growth change accounts for all strains within their 20% thermal performance breadth, rather than the more commonly applied 80%[49] and should therefore be considered a conservative estimate, as many strains at 20% of their $μ_{max}$ might not survive. For cyanobacteria, which comprise several strains predicted to undergo a range expansion and become viable in regions in which they were not previously, we also estimated the potential range extent using

our viability criteria under future thermal conditions. Trends were discerned by averaging group proportional growth change across each latitude.

**Reporting summary**. Further information on research design is available in the Nature Research Reporting Summary linked to this article.

## Data availability

Phytoplankton growth rates from this study (https://doi.org/10.26008/1912/bco-dmo.839696.1)[76], derived thermal capacities (https://doi.org/10.26008/1912/bco-dmo.839713.1)[77], and estimated thermal traits (https://doi.org/10.26008/1912/bco-dmo.839689.1)[78] have been made available through the Biological and Chemical Ocean Data Management Office (BCO-DMO). Source data are provided with this paper.

## Code availability

Code to reproduce this analyses is available on GitHub (https://github.com/sianderson/PFT_thermal_response) and in an online archive at Zenodo[79].

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

## Acknowledgements

The ensemble mean of Earth System models from the Coupled Model Inter-comparison Project phase 5 (CMIP5), presented in the Fifth Assessment Report of the Intergovernmental Panel on Climate Change[50], was provided by the Royal Netherlands Meteorological Institute (http://climexp.knmi.nl). This research was supported by NSF awards OCE-1638834 and OPP-1543245 (TAR) and NASA award 80NSSC17K0561 (SD). SIA was partially supported by NSF award OIA-1655221.

## Author contributions

S.I.A., A.D.B., S.C., S.D., and T.A.R. conceived of the study, developed the methodology, and wrote the manuscript. S.I.A. curated and analyzed the growth-temperature data, ran the projected growth model, and generated all figures and supporting materials for this study. T.A.R. acquired the necessary funding.

## Competing interests

The authors declare no competing interests.
