## [Peer Review File · Nature Communications]

Reviewer comments, initial -

Reviewers' comments:

Reviewer #1 (Remarks to the Author):

This manuscript is very well written, clear and concise. The meta-analysis compilation is worthy of this journal and the results and data are useful to move the field of earth system modeling forward and to expand our understanding of global warming effects in marine biology. This work suggests that we should probably revise the assumed Q10 of 1.88 and make distinctions for the different Phytoplankton Functional types whenever possible. Finding differences in Q10 was not a surprise but it had not been done and it was time that the Q10 parameter was revised.

Regarding the methodology, it is straightforward and builds upon previous work. Some of the data and methods are an expansion or continuation of a previous paper by Thomas et al, which has been highly cited. Including the modeling section and projections (with all its caveats) was also done elegantly by comparing two distinct time periods and avoiding patterns related to climate modes and natural variability.

Figures are appealing, self-explanatory, accurate. I appreciated the schematics included in the graphics such as the thermal performance metrics in figure 3.

Citations are appropriate and well chosen.

One important comment that I have is that the explanation for not including diazotrophs in the study should be at least mentioned in the main text and then explained fully in the methods section. It should not be relegated to the methods section only. In my opinion, this is a big gap to leave out and the effects of global warming on diazotrophs have important implications in biogeochemical cycling. I think that they should have included diazotrophs in their study even if they were to be considered as a separate PFT. It is interesting however that they did not consider the big diversity within the dinoflagellates, did they exclude mixotrophic species, for example?

Reviewer #2 (Remarks to the Author):

REVIEW Marine Phytoplankton Functional Types Exhibit Diverse Responses to Thermal Change by Anderson et al.

The paper reports and analyses thermal traits (such as Q10 values, growth maxima and thermal ranges) of four major marine phytoplankton types individually (diatoms, coccolithophores, cyanobacteria and dinoflagellates). The authors then estimate the potential future response of the four phytoplankton types to warmer temperatures alone (in isolation of changes in light and nutrient concentration etc) due to global warming.

Overall, I found the paper to be very well written, the results are clearly presented and thoroughly discussed, and all statements/conclusions are supported by the figures/results. I also think a detailed analysis of the individual temperature responses of different phytoplankton types is overdue and

highly significant as it is missing in current projections of primary production despite important implications for the response of marine phytoplankton to future global warming. I have only very few comments and I recommend a publication in Nature Communications after my comments below have been addressed.

Abstract:

L 19-21 "Here we assessed phytoplankton functional diversity in relation to temperature using empirically derived thermal reaction norms from four principal contributors to marine productivity: diatoms, dinoflagellates, cyanobacteria, and coccolithophores." - I find this sentence difficult to understand and wonder if it can be simplified for the broader audience you are aiming at. Is "phytoplankton functional diversity in relation to temperature" necessary or could it be "phytoplankton functional responses to temperature"? Also, I am not sure if the term "thermal reaction norms" is commonly understood, I would at least add an explanatory sentence somewhere in the main text. Anyway, I leave this up to the authors, there is certainly nothing wrong about this sentence.

Introduction:

Regarding the reason for looking at effects of temperature alone (lines 61—67): It's true that temperature plays a key role & there is more confidence in temperature projections, however the response to future nutrient limitation is widely cited as the main driver of future phytoplankton decline. I like the approach of this paper to analyze the response to temperature changes in isolation, but lines 61-67 somehow make it sound as if looking at temperature changes alone results in more reliable future phytoplankton projections, which I think is not true.

L 75 "the temperature range where growth rates were greater than 20% of μ_{max} " - How can the growth rates be greater than μ_{max} ? Do you mean smaller than? Or do I misunderstand something? Same in L116/117: "For example, at 20C, diatoms and coccolithophores have the greatest μ_{max} despite having the lowest Q10 values (Table 118 1, Fig. 2B)."

I think the term " μ_{max} " should be defined somewhere. To me, it means maximum growth rate under ideal conditions, but here it seems to be something else, otherwise "greatest μ_{max} " does not make sense. I think I understand though what the authors are saying, that at 20C diatoms and coccolithophores have higher growth rates despite lower Q10s. So maybe it should be just μ instead of μ_{max} ?

Discussion/Fig 4: I understand that the projected changes in growth are calculated using the median temperature response of individual strains. However, I often heard the argument that if there are e.g. several diatom species and the water warms such that some species are above their optimal temperature or even above their upper thermal limit, then other species would take over. So there would not be an overall net decrease in growth but a shift towards other species (or potentially a decrease in biodiversity?).

In that context it may be worth discussing that current biogeochemistry models never predict decreases in growth rate directly due to warmer temperatures (because of the above argument I think). However, if the negative temperature response suggested by the authors is currently missing

in global models, the already predicted decrease in phytoplankton due to nutrient stress could potentially be way more severe (assuming that these two effects simply add up).

L 414-416 The title of the Sherman reference is half-missing

I hope this helps! I really enjoyed reading the manuscript.

Reviewer #3 (Remarks to the Author):

This paper examines how marine phytoplankton functional types/groups respond to temperature, and explores the consequences of warming for the performance of these types. Specifically, it uses previously collected lab measurements to:

- A) Identify group differences in parameters describing thermal reaction norms at the species/strain level ($T_{\mu\text{equiv}}$, DGE and WT) and group level (Q_{10})
- B) Examine latitudinal variation in the species/strain-level parameters
- C) Examine how temperature change will affect the growth of different phytoplankton groups.

This is an important topic of study and I find the writing to be clear and the text and figures fairly straightforward. However, I believe that there are substantial weaknesses.

My main criticisms are:

- 1) Of the three objectives stated above, both A and B have been done previously in three papers that the authors cite here, with largely the same dataset, methods and results (the dataset from this study included ~20% more strains). Thomas et al. 2012 and 2016 showed latitudinal and functional type variation in thermal reaction norm parameters. Kremer et al. 2017 showed functional type differences in Q_{10} s.

The overlap with these three papers is fairly substantial. In the abstract, the authors state “*Contrary to the commonly applied Eppley formulations, our data suggest phytoplankton functional types may be characterized by different temperature coefficients (Q_{10}), growth maxima, and thermal ranges which would drive dissimilar responses to each degree of temperature change*”. These have all been established in Thomas et al. 2016 and Kremer et al. 2017, with the same dataset and methods. And to take a specific result, the authors state “*Our results differed from the well-established and widely used Eppley curve, which characterized the thermal sensitivity of phytoplankton growth rates (Q_{10}) as 1.88. We instead found the four functional types examined to have a collective Q_{10} of 1.46*”. However, Kremer et al. 2017 found a very similar collective Q_{10} of 1.53 and also identified functional group differences. There are small differences in the parameter values estimated in these two papers because of the small differences in dataset, but those differences are not stressed in the paper, so it is unclear if they are meaningful. If they are (which is entirely possible), I would encourage the authors to make this clear for readers.

Regarding these first two objectives, the main differences in this study are in the creation of several new metrics to characterise the thermal reaction norms: $T_{\mu\text{equiv}}$, DGE and WT. I am not persuaded this represents an advance, though. The thermal reaction norm literature is overflowing with metrics that characterise the skewed unimodal shape, and many of these substantially overlap with each other. The new metrics the authors propose also overlap with existing ones in interpretation, global patterns, and inferences drawn (e.g. tropical taxa being more sensitive to warming), so it’s not clear what we gain by using them instead of existing ones. And using new metrics makes comparisons with existing literature more difficult. Furthermore, in defining some of these metrics, the authors use the growth rate at mean environmental temperature (T_{hab}) as a reference; however, this is not necessarily a very valuable reference because it ignores temperature variation and Jensen’s inequality.

2) Objective C is novel in its application at the functional group level (previously, inferences have been drawn at the whole community level), and could perhaps be a useful focus for a paper. Unfortunately, this is the weakest part of the analysis.

(i) As the authors note, these projections ignore nutrients, light and every other factor which we know shapes competition between groups. I accept that this is an insurmountable challenge at present, but think more caution is needed when discussing this analysis and drawing conclusions.

(ii) Unfortunately, there is a larger problem. The authors conclude that groups will decline in growth rate based on the median response of the species within the group. But the median is irrelevant here! If we accept the same limitations that the authors have (only temperature-dependence, no light or nutrients), then the group-level response is mathematically determined by the fastest-growing species, not the median. Even if all other species go extinct, if one species is able to grow at the same rate, the group's growth rate will be maintained. This may be ecologically unrealistic, but I submit that it follows from the premises that the authors ask us to accept. If we wish for a more realistic scenario, we must grapple with competition – and the light- and nutrient-dependence that have been ignored in this analysis.

This cannot simply be fixed by updating the analysis based on the fastest growing species, however. Unlike the median, the maximum is highly sensitive to sample size and experimental conditions. So the probability of finding more extreme values if more species are measured in each group is not small. I find this to be a problem that cannot be fixed without large amounts of better data.

3) As a more minor comment, the exclusion of diazotrophs from cyanobacteria ought to be made more prominent in the text and figure captions. They are among the most influential cyanobacteria (and phytoplankton), and this important analytical decision is easily missed.

Response to Reviewers, Anderson et al. NCOMMS-20-35168

Reviewer #1 (Remarks to the Author):

This manuscript is very well written, clear and concise. The meta-analysis compilation is worthy of this journal and the results and data are useful to move the field of earth system modeling forward and to expand our understanding of global warming effects in marine biology. This work suggests that we should probably revise the assumed Q10 of 1.88 and make distinctions for the different Phytoplankton Functional types whenever possible. Finding differences in Q10 was not a surprise but it had not been done and it was time that the Q10 parameter was revised.

Regarding the methodology, it is straightforward and builds upon previous work. Some of the data and methods are an expansion or continuation of a previous paper by Thomas et al, which has been highly cited. Including the modeling section and projections (with all its caveats) was also done elegantly by comparing two distinct time periods and avoiding patterns related to climate modes and natural variability.

Figures are appealing, self-explanatory, accurate. I appreciated the schematics included in the graphics such as the thermal performance metrics in figure 3.

Citations are appropriate and well chosen.

One important comment that I have is that the explanation for not including diazotrophs in the study should be at least mentioned in the main text and then explained fully in the methods section. It should not be relegated to the methods section only. In my opinion, this is a big gap to leave out and the effects of global warming on diazotrophs have important implications in biogeochemical cycling. I think that they should have included diazotrophs in their study even if they were to be considered as a separate PFT. It is interesting however that they did not consider the big diversity within the dinoflagellates, did they exclude mixotrophic species, for example?

We thank you for your kind and constructive remarks. To address your concerns:

1. Of course, diazotrophs are important and we had analyzed them as part of our study but clearly, we did not sufficiently explain why they were removed from our final manuscript. We have included an explanation for our omission of the diazotrophs in the main text on lines 75-77, methods (lines 277-282), as well as a new extended figure 5. Briefly, while we would have liked to include an assessment of the ecologically relevant diazotrophs, the current thermal growth data is insufficient (just 7 thermal reaction norms and 144 data points total) to confidently fit a 99th quantile regression (see the very broad 95% confidence intervals in extended figure 5). We have included our exploratory analysis in extended figure 5. We also note that subsequent studies assessing diazotroph thermal traits would be of great value to the scientific community in lines 282-283.
2. Unfortunately, we were unable to fully consider the diversity within the dinoflagellates with the current dataset. As part of our extensive quality control of the data, we verified that all dinoflagellate data we utilized was based on experiments using growth medium comprised of inorganic nutrients, rather than prey species, and therefore were autotrophically grown. However, we cannot exclude the possibility that some species may have ingested some bacteria to augment their growth (Seong et al. 2006). We have added a note in line 283-286 that this analysis is predominantly representative of autotrophic dinoflagellate growth.

Reviewer #2 (Remarks to the Author):

The paper reports and analyses thermal traits (such as Q10 values, growth maxima and thermal ranges) of four major marine phytoplankton types individually (diatoms, coccolithophores, cyanobacteria and dinoflagellates). The authors then estimate the potential future response of the four phytoplankton types to warmer temperatures alone (in isolation of changes in light and nutrient concentration etc) due to global warming.

Overall, I found the paper to be very well written, the results are clearly presented and thoroughly discussed, and all statements/conclusions are supported by the figures/results. I also think a detailed analysis of the individual temperature responses of different phytoplankton types is overdue and highly significant as it is missing in current projections of primary production despite important implications for the response of marine phytoplankton to future global warming. I have only very few comments and I recommend a publication in Nature Communications after my comments below have been addressed.

Abstract:

L 19-21 "Here we assessed phytoplankton functional diversity in relation to temperature using empirically derived thermal reaction norms from four principal contributors to marine productivity: diatoms, dinoflagellates, cyanobacteria, and coccolithophores." - I find this sentence difficult to understand and wonder if it can be simplified for the broader audience you are aiming at. Is "phytoplankton functional diversity in relation to temperature" necessary or could it be "phytoplankton functional responses to temperature"? Also, I am not sure if the term "thermal reaction norms" is commonly understood, I would at least add an explanatory sentence somewhere in the main text. Anyway, I leave this up to the authors, there is certainly nothing wrong about this sentence.

We appreciate the comment and have changed the sentence in question to "Here we assessed phytoplankton functional responses to temperature using empirically derived thermal growth rates from four principal contributors to marine productivity: diatoms, dinoflagellates, cyanobacteria, and coccolithophores."

Introduction:

Regarding the reason for looking at effects of temperature alone (lines 61—67): It's true that temperature plays a key role & there is more confidence in temperature projections, however the response to future nutrient limitation is widely cited as the main driver of future phytoplankton decline. I like the approach of this paper to analyze the response to temperature changes in isolation, but lines 61-67 somehow make it sound as if looking at temperature changes alone results in more reliable future phytoplankton projections, which I think is not true.

Thank you for bringing this to our attention. It was not our intention to lessen the role that nutrients, or any other variable, will have in driving phytoplankton communities. We only meant to state there is greater confidence in how sea surface temperature may be altered in the future, as opposed to nutrients, which are also influenced by complex biotic processes. We have adjusted the text (lines 62-69) to address this.

L 75 "the temperature range where growth rates were greater than 20% of μ_{max} " - How can the growth rates be greater than μ_{max} ? Do you mean smaller than? Or do I misunderstand something?

Same in L116/117: "For example, at 20C, diatoms and coccolithophores have the greatest μ_{max} despite having the lowest Q10 values (Table 118 1, Fig. 2B)."

I think the term “ μ_{\max} ” should be defined somewhere. To me, it means maximum growth rate under ideal conditions, but here it seems to be something else, otherwise “greatest μ_{\max} ” does not make sense. I think I understand though what the authors are saying, that at 20C diatoms and coccolithophores have higher growth rates despite lower Q10s. So maybe it should be just μ instead of μ_{\max} ?

We have made several changes that should clarify our message. By 20% of μ_{\max} , we had meant the product of $(0.2 \cdot \mu_{\max})$. We have adjusted the wording to make this clearer (line 80-82) and the line now reads " temperature range where growth rates were at least 20% of the μ_{\max} for that strain". Additionally, we reference μ_{\max} both in terms of strain maximum growth rate, given as the high point on a thermal reaction norm, and for each PFT, given as a point along the exponential fitted function. We agree this may have been confusing. To clarify, we have differentiated between the two definitions with the qualifier, “modeled”, defined in lines 114-115 and 118-120. We have adjusted references to this term, such as in the legend of Figure 2B. We chose not to change our wording to growth rates in general, as diatom growth rates are not universally greater than those of coccolithophores, but the exponential curve characterizing the maximum growth rate with temperature (modeled μ_{\max}) is greater for the diatoms.

Discussion/Fig 4: I understand that the projected changes in growth are calculated using the median temperature response of individual strains. However, I often heard the argument that if there are e.g. several diatom species and the water warms such that some species are above their optimal temperature or even above their upper thermal limit, then other species would take over. So there would not be an overall net decrease in growth but a shift towards other species (or potentially a decrease in biodiversity?).

In that context it may be worth discussing that current biogeochemistry models never predict decreases in growth rate directly due to warmer temperatures (because of the above argument I think). However, if the negative temperature response suggested by the authors is currently missing in global models, the already predicted decrease in phytoplankton due to nutrient stress could potentially be way more severe (assuming that these two effects simply add up).

Some models use changes in growth rate as an intermediate step to make projections about diversity (e.g. Thomas et al., 2012), due to the reasons the reviewer speculated. However, we purposely chose to avoid discussing diversity quantitatively for two reasons: (1) conclusions are strongly influenced by both sample size and the organisms represented in the dataset and (2) we find this intermediate information regarding changes in growth to be particularly interesting and worth highlighting, as rates have biogeochemical implications. For example, in low-latitude regions, where all groups are estimated to have a reduction in growth, it may be beneficial to know by how much when making primary production projections. Still, we highlight several ways in which changes in growth may ultimately lead to a reshuffling, both within PFTs and among them (lines 203-205, 210-219, 221-224, 249-252).

In reference to current global models, most *have* indeed incorporated a growth-temperature response (e.g. Dunne et al. 2013; Moore et al. 2013). Importantly, many models utilize a constant Q_{10} across functional types, such as that characterized by Eppley, resulting in a homogeneous trend across PFTs. Our study is novel in its differentiation of the temperature response among PFTs. Incorporating these new Q_{10} 's and compounding projections with each group's unique nutrient requirements, as suggested, will ultimately lead to more robust and accurate projections for the future.

L 414-416 The title of the Sherman reference is half-missing

Thank you for catching this error. It has since been corrected.

I hope this helps! I really enjoyed reading the manuscript.

Reviewer #3

This paper examines how marine phytoplankton functional types/groups respond to temperature, and explores the consequences of warming for the performance of these types. Specifically, it uses previously collected lab measurements to:

- A) Identify group differences in parameters describing thermal reaction norms at the species/strain level ($T_{\mu\text{equiv}}$, DGE and WT) and group level (Q10)
- B) Examine latitudinal variation in the species/strain-level parameters
- C) Examine how temperature change will affect the growth of different phytoplankton groups.

This is an important topic of study and I find the writing to be clear and the text and figures fairly straightforward. However, I believe that there are substantial weaknesses.

My main criticisms are:

- 1) Of the three objectives stated above, both A and B have been done previously in three papers that the authors cite here, with largely the same dataset, methods and results (the dataset from this study included ~20% more strains). Thomas et al. 2012 and 2016 showed latitudinal and functional type variation in thermal reaction norm parameters. Kremer et al. 2017 showed functional type differences in Q10s.

The overlap with these three papers is fairly substantial. In the abstract, the authors state “Contrary to the commonly applied Eppley formulation⁵, our data suggest phytoplankton functional types may be characterized by different temperature coefficients (Q10), growth maxima, and thermal ranges which would drive dissimilar responses to each degree of temperature change”. These have all been established in Thomas et al. 2016 and Kremer et al. 2017, with the same dataset and methods. And to take a specific result, the authors state “Our results differed from the well-established and widely used Eppley curve, which characterized the thermal sensitivity of phytoplankton growth rates (Q10) as 1.88. We instead found the four functional types examined to have a collective Q10 of 1.46”. However, Kremer et al. 2017 found a very similar collective Q10 of 1.53 and also identified functional group differences. There are small differences in the parameter values estimated in these two papers because of the small differences in dataset, but those differences are not stressed in the paper, so it is unclear if they are meaningful. If they are (which is entirely possible), I would encourage the authors to make this clear for readers.

Regarding these first two objectives, the main differences in this study are in the creation of several new metrics to characterise the thermal reaction norms: $T_{\mu\text{equiv}}$, DGE and WT. I am not persuaded this represents an advance, though. The thermal reaction norm literature is overflowing with metrics that characterise the skewed unimodal shape, and many of these substantially overlap with each other. The new metrics the authors propose also overlap with existing ones in interpretation, global patterns, and inferences drawn (e.g. tropical taxa being more sensitive to warming), so it's not clear what we gain by using them instead of existing ones. And using new metrics makes comparisons with existing literature more difficult. Furthermore, in defining some of these metrics, the authors use the growth rate at mean environmental temperature (T_{hab}) as a reference; however, this is not necessarily a very valuable reference because it ignores temperature variation and Jensen's inequality.

We thank you for your comments as they suggest we did not adequately highlight the substantial differences between our study and those of others. We argue that none of

the aforementioned studies were repeated, but rather built upon. We have addressed each concern more thoroughly in the text and will review them here:

1. Briefly, Thomas et al. 2012 and 2016 conducted a substantial and valuable meta-analysis to discern trends in phytoplankton traits across latitudes. Our study built upon this body of work by relating those traits to a dynamic environment. For example, Thomas et al. examined the thermal niche (width of reaction norm), we examined the thermal range (span of temperatures where PFTs are viable) and use that to make biogeographic predictions for the future.
2. Contrary to the opinion of the reviewer, we used 2 well-established metrics, WT (warming tolerance) and TSM (thermal safety margin) (referenced on line 154-155 and in the methods (e.g. Deutsch et al. 2008; Sunday et al. 2014; Comte and Olden 2017; Pinsky et al. 2019)), to characterize each group's capacity for warming, or their habitat proximity to various traits (thermal optima and maxima). While we did formulate one new metric, the DGE (distance to growth equivalence), we felt its absence from the literature represented an oversight which should be addressed: in terms of growth rate, organisms have a greater tolerance for warming than is currently being accounted for (lines 162-166, 346-351). These analyses do not address the traits directly, but rather showcase the ecological implications of having traits positioned differentially along a thermal axis. We feel these complement and emphasize previous work, rather than repeating it. We have tried to better assert this (e.g. lines 174-176). As an aside, our decision to use growth rate at the T_{hab} was in following the well-established methodology for ectotherms e.g. Deutsch et al. 2008. While it may not capture the variability in the environment, it still provides a useful means for compare traits and the environment.
3. In regards to Kremer et al. 2017, there are significant and important distinctions with our study, which we have now highlighted more thoroughly in the text (lines 115-118) as well as in a new extended data table 1. Kremer et al. was novel in proposing PFT differences from the Eppley formulation; however, they did not propose a temperature-PFT interaction (different exponents for each group), resulting in a constant Q_{10} across functional types. This difference is instrumental as the Q_{10} is an input in earth system models to modulate phytoplankton growth (Dutkiewicz et al. 2013, 2015), satellite-based estimations of primary production (Behrenfeld et al. 2005), phytoplankton growth rate standardizations (Chen and Liu 2010), as well as many other ecological implementations. We cannot overstate how substantial this differentiation is between the two studies. We hope displaying the specific equations that result from each study in the new extended table 1 will help to clarify these dissimilarities.
4. We also compared our new PFT exponential curves (previously termed growth maxima in the abstract – now the modeled μ_{max} to address Reviewer 1's comment) in ecologically meaningful ways. For example, in lines 136-144, we propose diatom's higher modeled μ_{max} may be the mechanism of its r-strategist characterization, outgrowing all other PFT's when nutrients are replete.
5. Finally, we chose to focus our study solely on marine phytoplankton, rather than a combination of freshwater and marine as was done in these two previous studies. Again, our work builds upon that of previous work, and we strongly argue that it is not a repetition of past methodologies.

- 2) Objective C is novel in its application at the functional group level (previously, inferences have been drawn at the whole community level), and could perhaps be a useful focus for a paper. Unfortunately, this is the weakest part of the analysis.
- a. (i) As the authors note, these projections ignore nutrients, light and every other factor which we know shapes competition between groups. I accept that this is an insurmountable challenge at present, but think more caution is needed when discussing this analysis and drawing conclusions.
 - b. (ii) Unfortunately, there is a larger problem. The authors conclude that groups will decline in growth rate based on the median response of the species within the group. But the median is irrelevant here! If we accept the same limitations that the authors have (only temperature dependence, no light or nutrients), then the group-level response is mathematically determined by the fastest-growing species, not the median. Even if all other species go extinct, if one species is able to grow at the same rate, the group's growth rate will be maintained. This may be ecologically unrealistic, but I submit that it follows from the premises that the authors ask us to accept. If we wish for a more realistic scenario, we must grapple with competition – and the light- and nutrient-dependence that have been ignored in this analysis.

This cannot simply be fixed by updating the analysis based on the fastest growing species, however. Unlike the median, the maximum is highly sensitive to sample size and experimental conditions. So the probability of finding more extreme values if more species are measured in each group is not small. I find this to be a problem that cannot be fixed without large amounts of better data.

We thank you for your comments but argue below that the median growth is the best way to achieve the aims of the study. By illustrating the median proportional growth change, we aimed to convey (1) the physiological impact temperature would have on each PFT based on their unique thermal traits and dependencies, (2) the diversity reshuffling either among PFTs or within and (3) the intensity with which these both of these may occur. However, we recognize caveats are needed, as by definition, some strains will do better and some will fare worse than the results presented. We have added this to the methods, lines 370-373. We have also taken care to ensure that we are not overstating the conclusions drawn from this analysis, especially as they relate to competition (e.g. lines 203-205, 210-219, 249-252). The reviewer is correct in that the median may not accurately assess each PFT's competitive ability as a whole, and we have adjusted certain sections to reflect this (e.g. lines 221-224).

However, we argue that the median growth rate is the best method to convey our aforementioned aims. As stated, the median is not as sensitive to sample size or outliers as the maximum, yet it does convey how the majority of strains from each PFT will fare. By using a 20% thermal performance breadth, we have also conservatively estimated the median, as many strains that we included might not survive at 20% of their μ_{max} (lines 373-376). To clarify our aims, we have adjusted several parts of the “changing rates and shifting ranges” section (e.g. lines 184-187, 203-205, 210-219, 249-252) to reflect these changes.

- 3) As a more minor comment, the exclusion of diazotrophs from cyanobacteria ought to be made more prominent in the text and figure captions. They are among the most influential cyanobacteria (and phytoplankton), and this important analytical decision is easily missed.

Thank you for this comment. Reviewer 1 also picked up on the omission. Of course, diazotrophs are important and we had analyzed them as part of our study but clearly, we did not sufficiently explain why they were removed from our final manuscript. We have included an explanation for our omission of the diazotrophs in the main text on lines 75-77, methods (lines 277-282), as well as a new extended figure 5. Briefly, while we would have liked to include an assessment of the ecologically relevant diazotrophs, the current thermal growth data is insufficient (just 7 thermal reaction norms and 144 data points total) to confidently fit a 99th quantile regression (see the very broad 95% confidence intervals in extended figure 5). We have included our exploratory analysis in extended figure 5. We also note that subsequent studies assessing diazotroph thermal traits would be of great value to the scientific community in lines 282-283.

References:

- Behrenfeld, M. J., E. Boss, D. A. Siegel, and D. M. Shea. 2005. Carbon-based ocean productivity and phytoplankton physiology from space. *Global Biogeochem. Cycles* **19**: 1–14. doi:10.1029/2004GB002299
- Chen, B., and H. Liu. 2010. Relationships between phytoplankton growth and cell size in surface oceans: Interactive effects of temperature, nutrients, and grazing. *Limnol. Oceanogr.* **55**: 965–972. doi:10.4319/lo.2010.55.3.0965
- Comte, L., and J. D. Olden. 2017. Climatic vulnerability of the world's freshwater and marine fishes. *Nat. Clim. Chang.* **7**: 718–722. doi:10.1038/nclimate3382
- Deutsch, C. A., J. J. Tewksbury, R. B. Huey, K. S. Sheldon, C. K. Ghalambor, D. C. Haak, and P. R. Martin. 2008. Impacts of climate warming on terrestrial ectotherms across latitude. *Proc. Natl. Acad. Sci.* **105**: 6668–6672. doi:10.1073/pnas.0709472105
- Dunne, J. P., J. G. John, E. Shevliakova, and others. 2013. GFDL's ESM2 Global Coupled Climate-Carbon Earth System Models. Part II: Carbon System Formulation and Baseline Simulation Characteristics. *J. Clim.* **26**: 2247–2267. doi:10.1175/JCLI-D-12-00150.1
- Dutkiewicz, S., J. J. Morris, M. J. Follows, J. Scott, O. Levitan, S. T. Dyhrman, and I. Berman-Frank. 2015. Impact of ocean acidification on the structure of future phytoplankton communities. *Nat. Clim. Chang.* **5**: 1002–1006. doi:10.1038/nclimate2722
- Dutkiewicz, S., J. R. Scott, and M. J. Follows. 2013. Winners and losers: Ecological and biogeochemical changes in a warming ocean. *Global Biogeochem. Cycles* **27**: 463–477. doi:10.1002/gbc.20042
- Moore, J. K., K. Lindsay, S. C. Doney, M. C. Long, and K. Misumi. 2013. Marine Ecosystem Dynamics and Biogeochemical Cycling in the Community Earth System Model [CESM1(BGC)]: Comparison of the 1990s with the 2090s under the RCP4.5 and RCP8.5 Scenarios. *J. Clim.* **26**: 9291–9312. doi:10.1175/JCLI-D-12-00566.1
- Pinsky, M. L., A. M. Eikeset, D. J. McCauley, J. L. Payne, and J. M. Sunday. 2019. Greater vulnerability to warming of marine versus terrestrial ectotherms. *Nature* **569**: 108–111. doi:10.1038/s41586-019-1132-4
- Seong, K. A., H. J. Jeong, S. Kim, G. H. Kim, and J. H. Kang. 2006. Bacterivory by co-occurring red-tide algae, heterotrophic nanoflagellates, and ciliates. *Mar. Ecol. Prog. Ser.* **322**: 85–97. doi:10.3354/meps322085
- Sunday, J. M., A. E. Bates, M. R. Kearney, R. K. Colwell, N. K. Dulvy, J. T. Longino, and R. B. Huey. 2014. Thermal-safety margins and the necessity of thermoregulatory behavior across latitude and elevation. *Proc. Natl. Acad. Sci.* **111**: 5610–5615. doi:10.1073/pnas.1316145111

Reviewer comments, second round review-

Reviewer #1 (Remarks to the Author):

Thank you for addressing all the concerns raised by the reviewers. I do not have any further comments.

Reviewer #4 (Remarks to the Author):

Summary

This study presents analyses of an expanded database of the thermal reaction norms of marine phytoplankton, considering potential differences that may arise between major functional types, and the consequences of these differences for future phytoplankton communities as the world's oceans warm. It is noteworthy because of (1) its breadth, integrating fundamental analyses of functional/physiological data to broad scale projections of future ocean conditions, and (2) the clarity with which it is written, making it enjoyable and informative to read.

At this point, as a new reviewer of this m.s., I have only one substantial critique, and a number of smaller suggestions.

Major comment

The authors conclude, based on their analysis of an expanded data set of thermal reaction norms, that different phytoplankton functional types (PFTs, including diatoms, coccolithophores, dinoflagellates and cyanobacteria) exhibit different increases in maximum growth rate with temperature. This is an important claim, which, if verified, would motivate changes to global ecosystem models with ramifications for projections of the effects of climate change – as the authors themselves argue.

Significant claims require rigorous proof, and I am concerned that the current results have not been sufficiently substantiated in the manuscript – yet many scientists, including it appears other reviewers of this m.s., may be quick to accept this conclusion. I do believe, however, that the authors have the opportunity to perform additional analyses that may well resolve my concern.

(1) The authors hypothesize that each PFT has a unique Q10/exponential coefficient/activation energy characterizing the relationship between maximum growth rate and temperature. An appropriate null hypothesis is that all PFTs have the **same** temperature dependence.

In essence, my concern is that the authors' alternative hypothesis is never adequately tested against this null hypothesis. They analyze data on each PFT separately (e.g. line 328-330), running four individual regressions, and report only the point estimates for their slope coefficients (Table 1, Extended Data Table 1). These are unaccompanied by estimates of confidence intervals. While the authors do report bootstrapped confidence bands around their exponential regressions (Fig. 1 & 2), these incorporate covariation between the slope and intercept of their regressions and do not

present any direct way of testing their hypothesis. Exponential curves are also notoriously difficult to compare by eye – for example, curves with different intercepts and identical exponents may appear very different over the biologically relevant temperature range.

The most rigorous solution to this concern would be to pool all of the data across PFTs, and run two separate quantile regressions: (i) allow only main effects of temperature and PFT on growth rate, and (ii) allow for an interaction between temperature and PFT.

It is the significance of the interaction term (or the results of a model comparison between these two models) that would provide the strongest evidence in support of the authors' claim. Absent this information, it is impossible to judge by eye whether the bounding relationships shown in Fig. 1 actually imply different thermal sensitivities. Providing confidence intervals on the point estimates would help as well, although these are more likely to be mis-interpreted than the approach recommended above.

Conducting these analyses and presenting the full results would allow the scientific community to decide with confidence if this important claim can be rigorously supported.

(2) As a secondary note on this topic, and in reference to the exchange between reviewer 3 and the authors - it appears Kremer et al. 2017 L&O did consider the possibility that PFTs might exhibit different temperature sensitivity. They concluded that their assembled data was insufficient to test this hypothesis rigorously using quantile regression (page 6, right column; see also supplemental material). However, they did test this question using a different approach (page 7, left column), and did not find support for separate temperature sensitivities among PFTs.

Due to their efforts to expand the available data on phytoplankton thermal reaction norms, the authors of the current m.s. may well have overcome the data limitations that seem to have concerned Kremer et al., which would be exciting! However, it seems important to directly address this issue – are the authors concerned about whether the available data limits their power to detect limitations, and if not, why?

In either event, it also seems appropriate to explicitly discuss why (if substantiated) the authors' current conclusions differ from those of Kremer et al. Is it due to a difference in analytical approach? (note that the same analysis based on metabolic theory could be applied by the authors of the current m.s. to check this) Or is it the result of an expanded data set?

(3) I do want to recognize that other components of this paper (Fig. 3 and 4) are independent of this initial analysis, and so should be unaffected by the resolution of my above critiques.

Minor comments

L43-44 – just to be precise, most models do seem to account for a difference in growth rate among PFTs or size classes (e.g., in intercept) but not in the sensitivity of growth rate to temperature.

L50-51 – a very recent paper by Ward et al. might be of interest to the authors (<https://www.pnas.org/content/118/10/e2007388118.short>), suggesting that the dispersal ability of phytoplankton species is more limited than generally thought, because they must pass through unfavorable conditions and suffer the consequences even given some adaptive potential.

L82-84 – are the results consistent with what you might find if you were more selective about which curves to retain, but then work with T_{max} explicitly?

L85-87 – it's unlikely that all curve fits were left skewed; any individual fits in which the exponential coefficient from the Norberg curve is estimated to be negative would formally be right skewed. By eye, there are several of these (Fig. 1). Although maybe this conclusion is based on averages of data in Fig. 2A? I couldn't find an explicit statement for how these 'change in performance' values were calculated, but I probably missed something.

L90 – here and elsewhere throughout the m.s., please report quantitative estimates of the p-values. These can be critical to enabling future studies to build on your work (e.g., through meta-analyses). For very small values ($p = 1e-10$ or such) it's fine to truncate to $p < 0.0001$ or similar.

L91-92 – Speculatively, if there are consistent differences in the skewness of thermal reaction norms by PFT, then the use of the $\mu_{20\%}$ metric might influence this conclusion. For example, if cyanobacteria have more skewed reaction norms than other PFTs, their μ_{20} would be artificially narrower, especially at lower temperatures. I don't know that this is the case, but perhaps worth considering?

L94-95 – excellent, glad you tested this! Was there also any consideration of the geographic bias in sampling? (e.g., tropical taxa tend to be chronically undersampled).

L100 – I believe dinoflagellates also have vastly larger genomes than other taxa, which could depress their growth rates. I'm not sure I've heard convincing explanations for this yet, however.

L117 – see major comment, this was actually considered in Kremer et al. 2017

L118 – no apostrophe in coefficients?

L130 – what about controlling for differences due to cell size? This appears to be one significant factor leading to a departure from Eppley's values, at least in Kremer et al. 2017

L188 – perhaps 'premise' not 'preface'?

L221 – this conclusion might be especially sensitive to under sampling in the tropics

The reviewer comments are in black font and our responses are in blue.

REVIEWER COMMENTS

Reviewer #1 (Remarks to the Author):

Thank you for addressing all the concerns raised by the reviewers. I do not have any further comments.

We thank you for your time and consideration.

Reviewer #4 (Remarks to the Author):

Summary

This study presents analyses of an expanded database of the thermal reaction norms of marine phytoplankton, considering potential differences that may arise between major functional types, and the consequences of these differences for future phytoplankton communities as the world's oceans warm. It is noteworthy because of (1) its breadth, integrating fundamental analyses of functional/physiological data to broad scale projections of future ocean conditions, and (2) the clarity with which it is written, making it enjoyable and informative to read.

At this point, as a new reviewer of this m.s., I have only one substantial critique, and a number of smaller suggestions.

Major comment

The authors conclude, based on their analysis of an expanded data set of thermal reaction norms, that different phytoplankton functional types (PFTs, including diatoms, coccolithophores, dinoflagellates and cyanobacteria) exhibit different increases in maximum growth rate with temperature. This is an important claim, which, if verified, would motivate changes to global ecosystem models with ramifications for projections of the effects of climate change – as the authors themselves argue.

Significant claims require rigorous proof, and I am concerned that the current results have not been sufficiently substantiated in the manuscript – yet many scientists, including it appears other reviewers of this m.s., may be quick to accept this conclusion. I do believe, however, that the authors have the opportunity to perform additional analyses that may well resolve my concern.

(1) The authors hypothesize that each PFT has a unique Q10/exponential coefficient/activation energy characterizing the relationship between maximum growth rate and temperature. An appropriate null hypothesis is that all PFTs have the *same*

temperature dependence.

In essence, my concern is that the authors' alternative hypothesis is never adequately tested against this null hypothesis. They analyze data on each PFT separately (e.g. line 328-330), running four individual regressions, and report only the point estimates for their slope coefficients (Table 1, Extended Data Table 1). These are unaccompanied by estimates of confidence intervals. While the authors do report bootstrapped confidence bands around their exponential regressions (Fig. 1 & 2), these incorporate covariation between the slope and intercept of their regressions and do not present any direct way of testing their hypothesis. Exponential curves are also notoriously difficult to compare by eye – for example, curves with different intercepts and identical exponents may appear very different over the biologically relevant temperature range.

The most rigorous solution to this concern would be to pool all of the data across PFTs, and run two separate quantile regressions: (i) allow only main effects of temperature and PFT on growth rate, and (ii) allow for an interaction between temperature and PFT.

It is the significance of the interaction term (or the results of a model comparison between these two models) that would provide the strongest evidence in support of the authors' claim. Absent this information, it is impossible to judge by eye whether the bounding relationships shown in Fig. 1 actually imply different thermal sensitivities. Providing confidence intervals on the point estimates would help as well, although these are more likely to be mis-interpreted than the approach recommended above.

Conducting these analyses and presenting the full results would allow the scientific community to decide with confidence if this important claim can be rigorously supported.

We greatly appreciate this constructive feedback. We tested the null hypothesis that phytoplankton functional groups were characterized by equivalent thermal growth rates and growth maxima (μ_{max}), which would result in analogous temperature dependencies (lines 119-120). However, “we instead found PFTs to exhibit significantly different thermal growth rates (Kruskal-Wallis, $p < 0.0001$) and μ_{max} (Extended Data Table 1)” (lines 120-122; also added to methods: lines 347-351), supporting our decision to model the μ_{max} -temperature relationship separately for each PFT. This resulted in

“widely varying Q_{10} values and y-intercepts between PFTs, with cyanobacteria and dinoflagellates displaying significantly lower intercepts than either coccolithophores or diatoms ($\alpha = 0.05$; Fig. 1, Table 1, Extended Data Table 2)” (lines 137-139).

We thank you for noting that we had previously omitted these coefficient confidence bounds, as this omission was done in error and is necessary for evaluating curve differences. We have included the 95% confidence intervals for each PFT's curve coefficients in the new **Extended Data Table 2** as well as trait comparisons in **Extended Data Table 1** to better convey PFT variability.

Additionally, we substantiated our claim further by pooling the data, as suggested, and conducting a model comparison with and without a temperature-PFT interaction:

“We also compared two models of pooled thermal growth rates using the Akaike information criterion with correction for small sample size (AICc), which showed that including a temperature-PFT interaction resulted in a stronger model (Extended Data Table 4)” (lines 351-353).

Results from this analysis illustrate that including an interaction, and therefore differentiating between PFTs, provides a better thermal sensitivity model. We include a description of these additional analyses in the Methods section, **lines 347-353** and reported results in the new **Extended Data Table 4**.

Overall, these new analyses demonstrate that the temperature dependence differs between groups. We thank the reviewer for pushing us to improve our analyses.

(2) As a secondary note on this topic, and in reference to the exchange between reviewer 3 and the authors - it appears Kremer et al. 2017 L&O did consider the possibility that PFTs might exhibit different temperature sensitivity. They concluded that their assembled data was insufficient to test this hypothesis rigorously using quantile regression (page 6, right column; see also supplemental material). However, they did test this question using a different approach (page 7, left column), and did not find support for separate temperature sensitivities among PFTs.

Due to their efforts to expand the available data on phytoplankton thermal reaction norms, the authors of the current m.s. may well have overcome the data limitations that seem to have concerned Kremer et al., which would be exciting! However, it seems important to directly address this issue – are the authors concerned about whether the available data limits their power to detect limitations, and if not, why?

In either event, it also seems appropriate to explicitly discuss why (if substantiated) the authors’ current conclusions differ from those of Kremer et al. Is it due to a difference in analytical approach? (note that the same analysis based on metabolic theory could be applied by the authors of the current m.s. to check this) Or is it the result of an expanded data set?

We thank you for your comments regarding the study by Kremer et al. 2017. After a thorough review of the papers referenced by Kremer et al., as well as the broader literature, we are unsure how Kremer et al. reached the conclusion that they did not have adequate data. To our knowledge, a formal method for discerning adequate sample size (i.e. a multivariate power analysis for quantile regression) has not yet been established, and has only recently been conceived for univariate quantile regression (Yanuar 2018). We would also like to note that a paper published in the same year and same journal as Kremer et al. supported differentiating between PFTs when evaluating activation energies (Chen and Laws 2017). However, that study did not employ a 99th quantile regression, which we argue is superior in capturing the μ_{max} -temperature dependency:

“We utilized the 99th quantile regression method³², which characterizes phytoplankton growth at the 99th percentile which we believe serves as a stronger estimator of group μ_{max} than methods that assess rates at the 50th

percentile or those that use only a subset of the growth data in curve fitting” (lines 124-127).

Both Chen and Laws 2017 and Kremer et al. 2017 highlight the contentious nature of this topic, as well as the importance of reassessing Eppley (Eppley 1972), particularly for global ecosystem models assessing the impacts of climate change. We have added a full description of these methods and our deviations from them, including our reasons to avoid the metabolic theory of ecology (MTE), in the methods, **lines 333-346**:

“[The 99th quantile regression] analysis differs from the metabolic theory of ecology (MTE) and ordinary least squares (OLS) regression, both of which have been utilized to evaluate the μ_{max} -temperature relationship previously^{17,76}. Unlike the MTE, which incorporates only the μ_{max} from each species for curve fitting¹⁷, or the OLS regression, which uses only data below the thermal optima⁷⁷, the 99th quantile method utilizes all growth data available to assess thermal dependencies, providing more realism and greater statistical power. Additionally, the 99th quantile describes community growth rates at the 99th percentile, providing a better estimation of maximum growth rates than the OLS method, which fits to the log-transformed mean (proximate to the 50th quantile, median). The 99th quantile method is also less error prone than the MTE methodology, which incorporates cell size estimations, which are difficult to approximate in the phytoplankton, as cell size can vary significantly within species⁷⁸ and the size-scaling of growth is strongly dependent on temperature^{13,77}. Thus, incorporating species size estimates unassociated with thermal experiments introduces a margin of error which we thought beneficial to avoid.”

Returning to the broader question of whether these PFTs should be evaluated separately or not (major comment #1), we again assert our confidence in deciding to differentiate between phytoplankton functional groups based on our findings that thermal growth rates, μ_{max} , and curve coefficients significantly differed between groups:

“...we tested the null hypothesis that all functional groups are characterized by equivalent thermal growth rates and thermal traits. We instead found PFTs to exhibit significantly different thermal growth rates (Kruskal-Wallis, $p < 0.0001$) and μ_{max} (Extended Data Table 1)” (lines 119-122) and “widely varying Q_{10} values and y-intercepts between PFTs, with cyanobacteria and dinoflagellates displaying significantly lower intercepts than either coccolithophores or diatoms ($\alpha = 0.05$; Fig. 1, Table 1, Extended Data Table 2)” (lines 137-139).

Lastly, we note that fitting univariate quantile regressions (temperature ~ growth) has previously been carried out for sample sizes smaller than those from individual PFTs within our study (e.g. Bissinger et al. 2008; Stawiarski et al. 2016). We hope this explanation has alleviated your concerns. A table of methods employed by studies assessing phytoplankton thermal dependencies has also been added to the extended data (**Extended Data Table 5**).

(3) I do want to recognize that other components of this paper (Fig. 3 and 4) are independent of this initial analysis, and so should be unaffected by the resolution of my above critiques.

We thank you for your comments of support.

Minor comments

L43-44 – just to be precise, most models do seem to account for a difference in growth rate among PFTs or size classes (e.g., in intercept) but not in the sensitivity of growth rate to temperature.

We have adjusted the wording from “*thermal traits*” to “*thermal growth sensitivities*” to increase wording precision.

L50-51 – a very recent paper by Ward et al. might be of interest to the authors (<https://www.pnas.org/content/118/10/e2007388118.short>), suggesting that the dispersal ability of phytoplankton species is more limited than generally thought, because they must pass through unfavorable conditions and suffer the consequences even given some adaptive potential.

The modeling study by Ward et al. 2021 is compelling, suggesting some thermal gradients are too severe for phytoplankton to traverse, even with adaptation. However, empirical evidence suggests the contrary – that phytoplankton are able to disperse across the global ocean (Whittaker and Ryneerson, PNAS, 2017). Though the mechanisms that allow for unlimited dispersal and gene flow are unclear, we find this evidence of geographically unstructured diatom populations to be substantial, supporting our decision to ultimately model PFT growth in all thermally viable regions. However, we have included a citation of the Ward et al 2021 study, referencing that we formulated our projections following the unlimited dispersal hypothesis, as empirical evidence suggests, rather than the limited dispersal hypothesis as the Ward modeling study suggests:

“Assuming no limits to dispersal in phytoplankton, including those resulting from thermal boundaries, as supported by empirical observations⁵² rather than modeling simulations⁵³... we evaluated the growth of every strain at all thermally viable locations across the global ocean” (lines 203-208).

L82-84 – are the results consistent with what you might find if you were more selective about which curves to retain, but then work with T_{max} explicitly?

In this analysis, we sought to examine the viable thermal range of each phytoplankton strain, characterized from $\mu_{\max 20\%}$, our defined thermal minimum, to the $\mu_{\max 20\%}$ on the opposite side of the curve, our defined thermal maximum. Because this represents a range of temperatures for each isolate, a singular value like the T_{max} would not capture the full range of temperatures at which each PFT thrives.

L85-87 – it’s unlikely that all curve fits were left skewed; any individual fits in which the exponential coefficient from the Norberg curve is estimated to be negative would formally be right skewed. By eye, there are several of these (Fig. 1). Although maybe this conclusion is based on averages of data in Fig. 2A? I couldn’t find an explicit statement for how these ‘change in performance’ values were calculated, but I probably missed something.

You are correct in noting that several strains had right-skewed thermal reaction norms; however, we were referring to average trends and have adjusted our language to reflect this: “PFTs were characterized by negatively skewed thermal reaction norms on average...” (lines 99-100). We also thank you for noting that we had not fully described the ‘change in performance’ metric outside of the Figure 2 caption. We have added the following description to the main text (lines 85-89):

“Differences in the absolute change in performance were then evaluated for each strain by assessing the thermal reaction norm slope ascending to or descending from the μ_{max} to the $\mu_{20\%max}$ ($|\mu|/^{\circ}C$, Fig. 2A inset). Reaction norm slopes below and above the thermal optima (T_{opt} ; temperature at μ_{max}) were used to discern average PFT skewness, with a greater slope above the T_{opt} indicative of a negatively skewed curve (Fig. 2A).”

We hope these additions help to increase textual clarity.

L90 – here and elsewhere throughout the m.s., please report quantitative estimates of the p-values. These can be critical to enabling future studies to build on your work (e.g., through meta-analyses). For very small values ($p = 1e-10$ or such) it’s fine to truncate to $p < 0.0001$ or similar.

We have expanded all p-values in the text to 4 significant figures, and added a new **Extended Data Table 1** to display p-values from additional trait comparisons.

L91-92 – Speculatively, if there are consistent differences in the skewness of thermal reaction norms by PFT, then the use of the mu20% metric might influence this conclusion. For example, if cyanobacteria have more skewed reaction norms than other PFTs, their mu20 would be artificially narrower, especially at lower temperatures. I don’t know that this is the case, but perhaps worth considering?

In this study, we did not conduct a comparative analysis of skewness between PFTs. However, in a previous analysis, utilizing much of the same data, no differences in curve skewness were discerned across functional groups (Thomas et al. 2016). Thus, the estimated 20% thermal performance breadth should *not* vary as a result of differences in skewness. We have added a reference to this study in regards to curve skewness on **line 100**.

L94-95 – excellent, glad you tested this! Was there also any consideration of the geographic bias in sampling? (e.g., tropical taxa tend to be chronically undersampled).

While we recognize there are several under sampled regions within the dataset, both tropical and polar, we did not find any significant correlation between niche width and absolute latitude and have added this to **line 96-99**. We would love to see more phytoplankton growth data published, especially from chronically under-sampled regions (**lines 245-246**).

L100 – I believe dinoflagellates also have vastly larger genomes than other taxa, which could depress their growth rates. I’m not sure I’ve heard convincing explanations for this yet, however.

We thank you for bringing this to hypothesis to our attention. From what we could discern, the dinoflagellate genome-growth hypothesis (Tang 1996) remains unproven. If you are aware of a study argues otherwise, we would happily include it, though at this time we have omitted a reference to this hypothesis.

L117 – see major comment, this was actually considered in Kremer et al. 2017

We have added explanations on **lines 119-127 & 333-353**, as well as **Extended Data Table 1 & 2**, to provide support for this statement.

L118 – no apostrophe in coefficients?

Thank you for catching this error. The apostrophe has been removed.

L130 – what about controlling for differences due to cell size? This appears to be one significant factor leading to a departure from Eppley’s values, at least in Kremer et al. 2017

Though size can be a valuable factor to discern metabolic demand in macro-organisms, in the phytoplankton, it is not-straightforward. To highlight the challenges associated with incorporating cell size estimates, we have added the following text to our methods:

“The 99th quantile method is also less error prone than the MTE methodology, which incorporates cell size estimations, which are difficult to approximate in the phytoplankton, as cell size can vary significantly within species⁷⁸ and the size-scaling of growth is strongly dependent on temperature^{13,77}. Thus, incorporating species size estimates unassociated with thermal experiments introduces a margin of error which we thought beneficial to avoid” (**lines 341-346**).

Future thermal studies which include size measurements in their experimental design would prove beneficial to the scientific community, but until that data is available, we do not believe cell size is a valuable factor to include because it can change both during the life cycle (e.g. diatoms) and with environmental conditions (e.g. temperature).

L188 – perhaps ‘premise’ not ‘preface’?

'Preface' has been changed to 'premise.'

L221 – this conclusion might be especially sensitive to under sampling in the tropics

We thank you for noting that our projections may be affected by under sampling of the region. We have noted this in the main text on **lines 245-246**.

Reviewer comments, third round review-

Reviewer #4 (Remarks to the Author):

Summary

Thanks to the authors for their conscientious efforts to respond to my prior concerns and comments; the revised manuscript is much stronger. To recap: one of the most important and interesting results of this paper is the finding that temperature increases the maximum growth rate of different functional types of phytoplankton to varying degrees. This contrasts with the prevailing assumption in key models that a single value applies to all phytoplankton, and implies that warming will have differential effects on phytoplankton functional types (PFTs). In the previous draft, this conclusion was provided, but certain critical supporting results (such as the estimates and uncertainties of the temperature sensitivity of each PFT, and overall model comparisons) were lacking. In their revisions, the authors have provided this additional information. I am largely satisfied by the changes, although I have a few remaining issues to share below.

Major comment

- As reported by the authors, their results now support the conclusion that different PFTs have growth rates that scale with temperature differently (especially Extended Data Tables 2 & 4). These explicit quantitative results will also now help global modelers apply the authors' findings to future models if they so choose. This is a significant improvement, thank you!

However, I have some modest concerns that these results are not presented with sufficient nuance, given their potential importance. Two in particular may deserve reiterating in the text.

First, much of the overall conclusion that PFTs differ in their temperature sensitivity is driven by cyanobacteria (Extended Data Table 2). For example, the 95% CIs on the slope parameter in this table overlap for coccolithophores, diatoms, and dinoflagellates, and removing cyanobacteria from the analysis appears to dramatically decrease the AIC difference between models that allow or do not allow temperature sensitivity to differ by group (kudos to the authors for providing their code and data as part of their submission). I think this is noteworthy, it would be interesting to reflect on why cyanobacteria in particular seem to have a much steeper temperature sensitivity than the other PFTs. Additionally, in the context of how these new results differ from prior studies, based on Extended Data Table 3, I suspect the largest expansion of the Thomas et al. 2012 database was the addition of maybe 20-25% more cyanobacteria curves. Perhaps this provided additional power for detecting the current result, relative to Kremer et al. 2017?

Second, the results of extreme quantile regressions can be quite touchy and sensitive to which quantile is selected (Koenker 2005). For example, taking the supplied code on Github, selecting a model allowing different slope parameters for each PFT, and then estimating the value of these parameters for a range of quantiles ≥ 90 th yields the following plots:

<See attached file>

Code:

```
plot(summary(rq(ln.r~temperature*group,tau=seq(0.9,0.99,0.001),data=rates)),parm=c(6,7,8),ols=F)
```

Here, y-axis values show the difference between the estimated slope coefficient for diatoms (the PFT with the most underlying data) and the functional type in question (dinoflagellates, coccolithophores, cyanobacteria). Grey shows confidence bands. Any time this region exceeds or falls below the horizontal line at $y = 0$, the inference would be that the PFT in question differs in its thermal sensitivity from diatoms. Patterns generally match the results in Extended Data Table 2 (e.g., cyanobacteria have consistently, substantially larger values). However, you can see that some conclusions, such as whether diatoms and dinoflagellates differ, depend on which quantile you select (97th? 98th? 99th? 99.5th?). How confident are you really that these slight differences between quantiles are meaningful, given uncertainty in growth rate measurements, data extraction, etc., etc.?

I'm not saying that what the authors have done is wrong, just trying to highlight that the results are not overwhelmingly conclusive; extreme value problems are really difficult. This is the latest chapter on a pattern that has been marveled at and debated since Eppley's work in the 70s – ultimately, still more data and study is required.

Minor comments

- The term 'thermal growth rates' is vague/confusing (e.g., line 120, 121). I think what you're referring to is just the distribution of all the measured growth rates in your data set for a given PFT? Why call them thermal, especially if in this initial analysis you're not accounting for temperature at all?

- Line 337-338 – “the 99th quantile method utilizes all growth data available to assess thermal dependencies, providing more realism and greater statistical power. [than OLS/MTE]” while I am not a formally trained statistician, I think this assertion is misleading and quite possibly wrong. Yes, the quantile method uses a larger initial data set than these other methods, but given what it is being estimated (an extreme quantile!), a large portion of this data contributes very little to the actual results. I think you can make a stronger case for what you've done by focusing on how the quantile regression may tell you something *different* than these other methods then you can by asserting it is more realistic/powerful. Based on re-reading Kremer et al. 2017, there is background material in Supplement 6 relevant to this point, having to do with patterns of dispersion (see first paragraph and reference to Koenker 2005), more akin to your statement on line 339-341

- Line 341-343 – while it is true, as indicated by the references you cite, that cell sizes are variable and challenging to approximate, you have not shown that this leads to errors in MTE estimations. Rather, that is an assumption that you are making. Please consider revising this statement, or providing direct evidence. To be clear, my issue is not with the fact that you've made a judgement call about the approach you use, based on your opinions about the utility of cell size, as in line 344-346. Such decisions are ultimately necessary for advancing projects! But, it is best to avoid making untested claims in support of these decisions that unwary readers might accept as truths.

- Extended Data Table 1: this table would be more useful if the differences in effect sizes (indicating magnitude and directionality of effect) in addition to P values were reported.

- Extended Data Table 4: the code for this AIC comparison does not appear to be included in the Github repo, currently? Not a big issue.

- Response letter, pg 4: “Lastly, we note that fitting univariate quantile regressions (temperature ~ growth) has previously been carried out for sample sizes smaller than those from individual PFTs within our study” – this is not an especially strong argument in support of your case...? the fact that others have performed weaker analyses in the past relative to the current analysis does not itself guarantee that the current analysis is sufficient.

The reviewer comments are in black font and our responses are in blue.

Reviewer #4 (Remarks to the Author): Summary

Thanks to the authors for their conscientious efforts to respond to my prior concerns and comments; the revised manuscript is much stronger. To recap: one of the most important and interesting results of this paper is the finding that temperature increases the maximum growth rate of different functional types of phytoplankton to varying degrees. This contrasts with the prevailing assumption in key models that a single value applies to all phytoplankton, and implies that warming will have differential effects on phytoplankton functional types (PFTs). In the previous draft, this conclusion was provided, but certain critical supporting results (such as the estimates and uncertainties of the temperature sensitivity of each PFT, and overall model comparisons) were lacking. In their revisions, the authors have provided this additional information. I am largely satisfied by the changes, although I have a few remaining issues to share below.

Major comment

- As reported by the authors, their results now support the conclusion that different PFTs have growth rates that scale with temperature differently (especially Extended Data Tables 2 & 4). These explicit quantitative results will also now help global modelers apply the authors' findings to future models if they so choose. This is a significant improvement, thank you!

However, I have some modest concerns that these results are not presented with sufficient nuance, given their potential importance. Two in particular may deserve reiterating in the text.

First, much of the overall conclusion that PFTs differ in their temperature sensitivity is driven by cyanobacteria (Extended Data Table 2). For example, the 95% CIs on the slope parameter in this table overlap for coccolithophores, diatoms, and dinoflagellates, and removing cyanobacteria from the analysis appears to dramatically decrease the AIC difference between models that allow or do not allow temperature sensitivity to differ by group (kudos to the authors for providing their code and data as part of their submission). I think this is noteworthy, it would be interesting to reflect on why cyanobacteria in particular seem to have a much steeper temperature sensitivity than the other PFTs. Additionally, in the context of how these new results differ from prior studies, based on Extended Data Table 3, I suspect the largest expansion of the Thomas et al. 2012 database was the addition of maybe 20-25% more cyanobacteria curves. Perhaps this provided additional power for detecting the current result, relative to Kremer et al. 2017?

We thank you for highlighting a need to better convey the impact of cyanobacteria on subsequent analyses. To the results, we have described how cyanobacteria differed in their exponential slope:

“Cyanobacteria were also characterized by a significantly greater exponential slope than all other PFTs ($\alpha=0.05$; Fig. 1, Table 1, Supplementary Table 3), which resulted in a higher Q_{10} of 2.13, relative to Eppley, and contributed to the observed PFT-temperature interaction (Supplementary Table 4).” (lines 184-187).

Though an explanation for these differences has not yet been established, we note that these groups “represent distinct phyla with complex evolutionary histories”, which may attribute to the

observed physiological differences (lines 218-221). We have also noted that our dataset differed from those previous, which had fewer marine, non- diazotrophic cyanobacteria strains:

“These modifications resulted in PFT compilations that differed by as much as 72% (23 of 32 cyanobacteria strains; Supplementary Table 1) from previous work^{16,17”} (lines 101-103).

We hope these textual modifications help to highlight that cyanobacteria differed significantly from the other groups examined, contributing to the PFT-temperature relationship, as well as some explanation for these observations.

Second, the results of extreme quantile regressions can be quite touchy and sensitive to which quantile is selected (Koenker 2005). For example, taking the supplied code on Github, selecting a model allowing different slope parameters for each PFT, and then estimating the value of these parameters for a range of quantiles ≥ 90 th yields the following plots:

<See attached file>

Code:

```
plot(summary(rq(ln.r~temperature*group,tau=seq(0.9,0.99,0.001),data=rates)),parm=c(6,7,8),ols=F)
```

Here, y-axis values show the difference between the estimated slope coefficient for diatoms (the PFT with the most underlying data) and the functional type in question (dinoflagellates, coccolithophores, cyanobacteria). Grey shows confidence bands. Any time this region exceeds or falls below the horizontal line at $y = 0$, the inference would be that the PFT in question differs in its thermal sensitivity from diatoms. Patterns generally match the results in Extended Data Table 2 (e.g., cyanobacteria have consistently, substantially larger values). However, you can see that some conclusions, such as whether diatoms and dinoflagellates differ, depend on which quantile you select (97th? 98th? 99th? 99.5th?). How confident are you really that these slight differences between quantiles are meaningful, given uncertainty in growth rate measurements, data extraction, etc., etc.?

I'm not saying that what the authors have done is wrong, just trying to highlight that the results are not overwhelmingly conclusive; extreme value problems are really difficult. This is the latest chapter on a pattern that has been marveled at and debated since Eppley's work in the 70s – ultimately, still more data and study is required.

We thank you for both noting the sensitive nature of quantile regressions and for taking the time to illustrate your point, as it aided in our ability to address this concern. We have added more discussion of quantile regression to our methods (lines 406-410):

“Though results from quantile regression can be sensitive to the selected quantile (here 99th) and require substantial data inputs to estimate, we found this method preferential to other methodologies previously utilized to evaluate the μ_{max} -temperature relationship^{17,76}, such as the metabolic theory of ecology (MTE) or ordinary least squares (OLS) regression.”

We hope this textual note highlights some of the caveats with using quantile regression.

Minor comments

- The term 'thermal growth rates' is vague/confusing (e.g., line 120, 121). I think what you're referring to is just the distribution of all the measured growth rates in your data set for a given PFT? Why call them thermal, especially if in this initial analysis you're not accounting for temperature at all?

We thank you for noting how "thermal" may lead to confusion. We had originally included this descriptor because measured rates were collected in temperature-growth studies, but have now omitted it in this context.

- Line 337-338 – "the 99th quantile method utilizes all growth data available to assess thermal dependencies, providing more realism and greater statistical power. [than OLS/MTE]" while I am not a formally trained statistician, I think this assertion is misleading and quite possibly wrong. Yes, the quantile method uses a larger initial data set than these other methods, but given what it is being estimated (an extreme quantile!), a large portion of this data contributes very little to the actual results. I think you can make a stronger case for what you've done by focusing on how the quantile regression may tell you something *different* than these other methods than you can by asserting it is more realistic/powerful. Based on re-reading Kremer et al. 2017, there is background material in Supplement 6 relevant to this point, having to do with patterns of dispersion (see first paragraph and reference to Koenker 2005), more akin to your statement on line 339-341

We appreciate you noting how our text may be misleading. We have adjusted our wording, eliminating that utilizing more data may provide greater statistical power. We have also taken your advice and referred to Koenker (2005) (**lines 415-417**):

"Evaluating an extreme quantile also enables quantile regression to better capture changes in growth rate dispersion and variation in response to temperature, when compared to OLS regression"⁷⁸

- Line 341-343 – while it is true, as indicated by the references you cite, that cell sizes are variable and challenging to approximate, you have not shown that this leads to errors in MTE estimations. Rather, that is an assumption that you are making. Please consider revising this statement, or providing direct evidence. To be clear, my issue is not with the fact that you've made a judgement call about the approach you use, based on your opinions about the utility of cell size, as in line 344-346. Such decisions are ultimately necessary for advancing projects! But, it is best to avoid making untested claims in support of these decisions that unwary readers might accept as truths.

We have adjusted our wording as to not imply that the MTE methodology was error prone, but rather that it required additional assumptions, which we opted to avoid:

*"Lastly, the 99th quantile method differs from MTE methodology in that it does not require assumptions regarding phytoplankton cell size, which is difficult to approximate in the phytoplankton, as it can vary significantly within species⁷⁹ and the size-scaling of growth is strongly dependent on temperature^{13,77}. Thus, we decided against incorporating species size estimates that were unassociated with thermal growth experiments" (**lines 417-422**).*

- Extended Data Table 1: this table would be more useful if the differences in effect sizes (indicating magnitude and directionality of effect) in addition to P values were reported.

To our knowledge, there is not a direct way to compute the effect size following a Dunn's post-hoc test. However, we have added the effect size following the Kruskal-Wallis tests performed on the μ_{\max} (**line 160**: $\epsilon^2 = 0.42$ [0.34, 0.50]) and on the growth rates between groups (**line 159**: $\epsilon^2 = 0.28$ [0.25, 0.31]). To Extended Data Table 1, we have also added the z-test statistic, which describes the difference in group means divided by their standard deviation. This can be used as a metric for understanding how variable the groups were in their thermal niche and growth rates. We hope these additions provide greater context for our results.

- Extended Data Table 4: the code for this AIC comparison does not appear to be included in the Github repo, currently? Not a big issue.

The AIC model comparison has been added to "Curve_comparison.R" on the GitHub repository, which has now been archived on Zenodo (<https://doi.org/10.5281/zenodo.5507532>).

- Response letter, pg 4: "Lastly, we note that fitting univariate quantile regressions (temperature ~ growth) has previously been carried out for sample sizes smaller than those from individual PFTs within our study" – this is not an especially strong argument in support of your case...? the fact that others have performed weaker analyses in the past relative to the current analysis does not itself guarantee that the current analysis is sufficient.

Thank you for this comment. We hope that our other arguments sufficiently explained our methodology.